# DISCRETIZED QUADRATIC INTEGRATE-AND-FIRE NEURON MODEL FOR DIRECT TRAINING OF SPIKING NEURAL NETWORKS

## ABSTRACT

Spiking Neural Networks (SNNs) are a promising alternative to traditional artificial neural networks, offering significant energy-saving potential. Conventional SNN approaches typically utilize the Leaky Integrate-and-Fire (LIF) neuron model, where voltage decays linearly, decreasing proportionally to its current value. However, this linear decay can inadvertently increase energy consumption and reduce model performance due to extraneous spiking activity. To address these limitations, we introduce the discretized Quadratic Integrate-and-Fire (QIF) neuron model, which applies a non-linear transformation to the voltage proportional to its magnitude. The QIF neuron model achieves substantial energy reductions, ranging from $1.43 - 4.21\times$ compared to the LIF neuron model. On static datasets (CIFAR-10, CIFAR-100) and neuromorphic datasets (CIFAR-10 DVS, N-Caltech-101, N-Cars, DVS128-Gesture), the QIF neuron model demonstrates competitive performance and improved accuracy over state-of-the-art results. Furthermore, the QIF neuron model produces smoother loss landscapes and larger local minima, leading to faster training convergence. Our findings suggest that the QIF neuron model offers a promising alternative to the widely adopted LIF neuron model.

## 1 INTRODUCTION

Artificial Neural Networks (ANNs) have seen mainstream adoption in recent years thanks to their success in domains from computer vision (Chai et al., 2021) to natural language processing (Khan et al., 2023). However, the energy demands of ANNs continue to grow (Yamazaki et al., 2022). In contrast, Spiking Neural Networks (SNNs) have gained attention as a more energy-efficient alternative. Unlike traditional ANNs, which synchronously process continuous-valued data, SNNs operate asynchronously on discrete events known as spikes. These spikes, driven by biologically inspired neuron dynamics, allow SNNs to replicate the brain's sparse connectivity and energy-efficient structure (Gerstner et al., 2014). As a result, when SNNs are implemented on hardware tailored to these characteristics, they have the potential to operate with lower energy consumption than traditional ANN models (Rathi & Roy, 2023). This type of hardware is typically called neuromorphic hardware. Examples of SNNs implemented on neuromorphic hardware can be seen from always-on speech recognition for edge devices (Tsai et al., 2017), using IBM TrueNorth (Akopyan et al., 2015), and ultra-low-power image classification (Lenz et al., 2023), using Intel Loihi 2 (Intel, 2021).

In the context of deep learning, Wu et al. (2018) introduced the widely adopted neuron model by discretizing the Leaky Integrate-and-Fire (LIF) neuron model Gerstner et al. (2014). Despite its popularity, the LIF model's dynamics are limited to linear decay proportional to its voltage. The impact of this linear decay on energy consumption, performance, and convergence speed has yet to be studied. Therefore, in this work, we

propose the discretized Quadratic Integrate-and-Fire (QIF) neuron model for deep spiking neural networks. Unlike the LIF neuron model, the QIF neuron model incorporates non-linear decay and growth dynamics that scale with the magnitude of the neuron's voltage. Our QIF neuron model is compared to recent approaches on static datasets such as CIFAR-10 and CIFAR-100, as well as on neuromorphic datasets, including CIFAR-10 DVS, N-Caltech-101, N-Cars, and DVS128-Gesture. Furthermore, we compare the LIF and QIF neuron models, analyzing their energy efficiency, accuracy, loss landscapes, training performance, and robustness to hyperparameter selection. To summarize, the contributions of our work are as follows:

- We introduce a discretized Quadratic Integrate-and-Fire neuron model for deep learning applications which showcases $1.43 - 4.21\times$ better energy efficiency than the LIF neuron model.
- We derive and prove an analytical equation for calculating surrogate gradient windows directly from the QIF neuron model parameters, minimizing the risk of naïve initialization and significant gradient mismatch during training.
- Our QIF neuron model is compared to recent state-of-the-art approaches, demonstrating competitive performance on static datasets and improved accuracy on several neuromorphic datasets. Additionally, our analysis reveals the QIF neuron model can exhibit smoother loss landscapes, larger local minima, and greater robustness to hyperparameter selection, resulting in faster convergence and superior performance compared to the LIF neuron model.

## 2 RELATED WORK

### 2.1 DEEP LEARNING WITH SPIKING NEURAL NETWORKS

In recent years, two training techniques have stood out when training deep-spiking neural networks. ANN-to-SNN conversion was the first training technique to show promising and competitive performance for SNNs. Typically, these works first train a traditional ANN that utilizes the ReLU activation function (Cao et al., 2015). The ANN then has all activation functions replaced with a spiking neuron model (Ding et al., 2021). Then, the threshold for each layer of neurons is adjusted to approximate the ReLU function. Recent works use approaches such as modifying the ReLU function to better match the dynamics of an SNN (Li et al., 2021a; Wang et al., 2023; Bu et al., 2022), incorporating learnable parameters into the ReLU function (Ho & Chang, 2021; Ding et al., 2021), and developing new SNN neuron models to better fit the ReLU structure (Gao et al., 2023). The main disadvantage of conversion techniques is their inability to utilize temporal dynamics and require many timesteps to achieve high accuracy (Duan et al., 2022).

Direct training with backpropagation can also be used with SNNs. Several techniques have been developed to overcome the non-differentiability of spikes (Yi et al., 2023). One of the most well-adopted techniques is surrogate gradients. Surrogate gradients attempt to approximate the derivative of the Heaviside function (a common function used to obtain spiking behavior) with respect to the membrane potential using a differentiable function (Wu et al., 2018). In addition to surrogate gradients, works employ various techniques to improve direct training performance. Some of these techniques include batch or membrane potential normalization (Zheng et al., 2021; Duan et al., 2022; Guo et al., 2023), developing new loss functions (Deng et al., 2022; Guo et al., 2022), and learning surrogate gradient behavior (Li et al., 2021b; Lian et al., 2023; Deng et al., 2023). Due to the lack of support for training on neuromorphic datasets when using ANN-to-SNN conversion techniques, we restrict any comparisons to direct training techniques.

### 2.2 NEURON MODELS AND PARAMETER LEARNING

When using direct training techniques, a few works such as Fang et al. (2021); Yao et al. (2022); Rathi & Roy (2023); Lian et al. (2023; 2024) make modifications to the Leaky Integrate-and-Fire (LIF) neuron model by either changing its dynamics or incorporating learnable parameters. Fang et al. (2021) propose a

learnable decay factor for the LIF neuron model, which can be independently optimized for each layer. Rathi & Roy (2023) takes this a step further by co-optimizing the decay and threshold of each spiking layer. Yao et al. (2022) proposed gating features, similar to long-short term memory, that can choose between various biological features implemented in their model. Lian et al. (2023) propose using a learnable decay parameter to dynamically adjust the surrogate gradient window to fit the LIF neuron's voltage distribution throughout the training process. Lian et al. (2024) proposes using a temporal-wise attention mechanism to selectively establish connections between current and past temporal data.

While the LIF neuron model has seen various improvements and has showcased promising performance in many deep-learning applications, its dynamics are fundamentally constrained to linear decay proportional to the neuron's voltage. The effect of these linear dynamics on the energy efficiency, model accuracy, and convergence of deep spiking neural networks remains unknown. Inspired by this, we look towards other neuron models to quantify and address this limitation.

## 3 BACKGROUND

### 3.1 SPIKING NEURAL NETWORKS

While ANNs use continuous-valued data to transmit information, SNNs use discrete events called spikes. In modern deep SNN research, the Leaky Integrate-and-Fire (LIF) Gerstner et al. (2014) neuron model is the most widely adopted, with its dynamics governed by

$$\tau \frac{du}{dt} = u_{rest} - u + RI, \tag{1}$$

where $\tau$ is a membrane time constant, $u$ is the membrane potential, $u_{rest}$ is the resting potential, $R$ is a linear resistor, and $I$ is pre-synaptic input. When using the LIF neuron in deep learning scenarios, discretization is required Duan et al. (2022). The most commonly used discretization was introduced by Wu et al. (2018), who utilized Euler's method to solve Equation 1. They defined their model as

$$u(t + 1) = \beta u(t) + I(t). \tag{2}$$

In Equation 2, $t$ denotes the current timestep, $\beta$ is a membrane potential decay factor, $u$ is the membrane potential of a neuron, and $I$ are pre-synaptic inputs into the neuron. Given a threshold, $u_{th}$, when $u(t) > u_{th}$, a spike is produced and is denoted $o(t + 1)$. Wu et al. (2018) further define an iterative update rule for both spatial and temporal domains as

$$u(t + 1) = \beta u(t)(1 - o(t - 1)) + I(t) \tag{3}$$
$$o(t + 1) = \Theta(u(t + 1) - u_{th}), \tag{4}$$

where $\Theta$ is the Heaviside function with $\Theta(x) = 0$ if $x < 0$, else $\Theta(x) = 1$. Equations 3 and 4 allow for forward and backward backpropagation to be implemented in both the spatial and temporal domains automatically using modern deep learning frameworks Zheng et al. (2021).

Both the ordinary differential equation, shown in Equation 1, and the discretized equation, shown in Equation 2, of the LIF neuron model are constrained to a linear decay directly proportional to the voltage.

### 3.2 SURROGATE GRADIENTS

One challenge with spiking neural networks is that the Heaviside function, $\Theta$, is not suitable for backpropagation-based training as its derivative is either undefined or 0. To overcome this issue, Wu et al. (2018) proposed using the derivative of an approximation to the Heaviside function with useful gradient information. This technique is called a surrogate gradient. One of the most popular surrogate gradient

functions is the rectangle function Zheng et al. (2021); Deng et al. (2022); Lian et al. (2023) and is defined by

$$\frac{\partial o_n(t)}{\partial u_n(t)} \approx \frac{1}{\alpha} sign(|u_n(t) - u_{th}| < \frac{\alpha}{2}). \tag{5}$$

$\alpha$ determines the width and area of the surrogate gradient and typically remains constant throughout training. The choice of $\alpha$ greatly affects the learning process of SNNs, with improper choices leading to gradient mismatch and approximation errors.

### 3.3 THRESHOLD-DEPENDENT BATCH NORMALIZATION

Ioffe & Szegedy (2015) first introduced the concept of batch normalization for ANNs to accelerate the training process by reducing the internal covariant shift of each layer. Batch normalization was only designed to normalize spatial data, not spatial-temporal data. On this note, Zheng et al. (2021) proposed threshold-dependent Batch Normalization (tdBN) which works by normalizing the channels of pre-synaptic input, $I$, in both the spatial and temporal domains based on the neuron's threshold, $u_{th}$. Suppose $I_k(t)$ represents the $k_{th}$ feature map of $I$ at timestep $t$. Then, we normalize each feature map $I_k = (I_k(1), I_k(2), \ldots, I_k(T))$ in the temporal domain by

$$\hat{I}_k = \frac{\eta u_{th}(I_k - \mathbb{E}[I_k])}{\sqrt{Var(I_k) + \epsilon}} \tag{6}$$

$$\bar{I}_k = \gamma \hat{I}_k + \xi, \tag{7}$$

where $\mathbb{E}$ and $Var$ compute the mean and variance of $I_k$ in the channel dimension, $\eta$ is used for residual connections, and $\gamma$ and $\xi$ are learnable parameters. Following tdBN, $I$ satisfies $I \sim \mathcal{N}(0, u_{th}^2)$.

### 3.4 TRAINING SPIKING NEURAL NETWORKS

We adopt the Spatial-Temporal Back Propagation (STBP) algorithm and training procedure described by Wu et al. (2018) to train our network. First, we infer our model on temporal data for $T$ timesteps. Then, similarly to Lian et al. (2023), to decode the model's output, we turn off the firing behavior of the final output neurons and accumulate their voltage over time as follows

$$u_i = \frac{1}{T} \sum_{t=1}^{T} W_{n-1}^{(i)} o_{n-1}(t), \quad i \in \{1, 2, \ldots, c\}, \tag{8}$$

where $c$ is the number of neurons in the output layer, $W$ is a weight matrix, and $o_{n-1}(t)$ are spikes from the previous layer. The element, $u_i$, with the largest value, is the predicted class. Using our output vector $u = (u_1, u_2, \ldots, u_c)$ and a label vector $y = (y_1, y_2, \ldots, y_c)$, we compute the cross entropy loss, $L$, between $u$ and $y$. Then, using the STBP algorithm and surrogate gradients, we can train our network. As described by Guo et al. (2023), we use the chain rule to update weights by

$$\frac{\partial L}{\partial W_n} = \sum_{t=1}^{T} \left( \frac{\partial L}{\partial o_n(t)} \frac{\partial o_n(t)}{\partial u_n(t)} + \frac{\partial L}{\partial u_n(t+1)} \frac{\partial u_n(t+1)}{\partial u_n(t)} \right) \frac{\partial u_n(t)}{\partial W_n}, \tag{9}$$

where $n$ is the layer of the network. In the above equation, $\frac{\partial o_n(t)}{\partial u_n(t)}$ is replaced with a surrogate gradient, such as the one seen in Equation 5.

# 4 METHOD

## 4.1 QUADRATIC INTEGRATE-AND-FIRE NEURON MODEL

The Hodkin-Huxley (HH) neuron model was created to mimic the activity of neurons found within a giant squid and has proven itself invaluable in the field of neuroscience (Gerstner et al., 2014). Over the years, simplifications of the HH neuron model have been introduced to reduce the computational complexity of its various equations and non-linear dynamics. The LIF neuron model is an extreme simplification that has proven itself to be a computationally efficient alternative. However, the LIF neuron model does not contain non-linear dynamics dependent on voltage as seen in the HH neuron model. We aim to bridge this gap by looking at other neuron models that contain non-linear dynamics without introducing large computational overhead. This initially led us to the Exponential Integrate-and-Fire (ExLIF) neuron model (Gerstner et al., 2014). The ExLIF neuron model simplifies the HH neuron model and maintains much of its non-linear dynamics. However, due to the large computational cost of the ExLIF neuron, an approximation called the Quadratic Integrate-and-Fire (QIF) neuron model is often used in experimental settings (Gerstner et al., 2014). Therefore, we examine the QIF neuron model as a promising alternative to the LIF neuron model. The QIF neuron model is defined by

$$\tau \frac{du}{dt} = a(u - u_{rest})(u - u_c) + RI,\tag{10}$$

where $\tau$ is a membrane time constant, $a$ is a sharpness parameter controlling the rate of decay, $u$ is the membrane potential, $u_{rest}$ is the resting potential, $u_c$ is the critical spiking threshold, $R$ is a resistor, and $I$ is the pre-synaptic input. Additionally, it must hold that $a > 0$ and $u_{rest} < u_c$. Unlike the LIF neuron model, the QIF neuron model contains non-linear voltage dynamics which are proportional to the square of the voltage. This allows the QIF neuron to have varying dynamics based on the neuron's current voltage. For example, the QIF neuron can decay rapidly when $u < u_{th}$, or increase rapidly, as $u$ approaches and exceeds $u_c$ (Gerstner et al., 2014).

As with the LIF model, the QIF model requires discretization for usage in a deep learning setting (Duan et al., 2022). Therefore, we introduce our discretized QIF neuron model, defined as

$$u(t + 1) = a(u(t) - u_{rest})(u(t) - u_c) + I(t)\tag{11}$$

where $u(t)$ and $I(t)$ are the membrane potential and pre-synaptic input at timestep $t$ with all other parameters and constraints following that of Equation 10. Details on the discretization can be found in Appendix C. When incorporating this neuron model into existing deep spiking neural network architectures, we adopt and modify the iterative update rule proposed by Wu et al. (2018) to obtain

$$I_n(t) = W_{n-1} \circ o_{n-1}(t)\tag{12}$$
$$u_n(t + 1) = a(u_n(t) - u_{rest})(u_n(t) - u_c) + I_n(t)\tag{13}$$
$$o_n(t + 1) = \Theta(u_n(t + 1) - u_{th})\tag{14}$$
$$u_n(t + 1) = u_n(t + 1)(1 - o_n(t + 1)) + u_{rest}o_n(t + 1).\tag{15}$$

In the above equation, $t$ denotes the timestep, $n$ denotes the layer of the network, $\circ$ denotes either matrix multiplication or convolution between a synaptic weight $w$ and spikes $o$, $I$ is pre-synaptic input, $u$ is the membrane potential, $u_{th}$ is the firing threshold, and $\Theta$ is the Heaviside function. When the membrane potential exceeds the firing threshold $u_{th}$, a spike will be produced, and its potential will be reset to $u_{rest}$.

## 4.2 SURROGATE GRADIENT WINDOW

When using a surrogate gradient with the LIF model, like in equation 5, a common assumption is that the voltage distribution is mean centered around zero. However, the quadratic dynamics of the QIF neuron model

usually do not conform to this assumption. Instead, the QIF neuron produces a voltage distribution with a non-zero mean and a variance that can widely change based on the chosen neuron parameter set. Therefore, determining an appropriate surrogate gradient window for the QIF neuron model can be challenging. To alleviate this issue, we derive a surrogate gradient window based on the statistical properties of our neuron model when the pre-synaptic input $I$ has been normalized with the tdBN technique in Equation 6. Assuming that during forward propagation, all pre-synaptic input is normalized with tdBN such that $I \sim \mathcal{N}(0, u_{th}^2)$, we propose Theorem 1 to explain the statistical properties of the QIF neuron model.

**Theorem 1.** *Under the discrete QIF neuron model using tdBN to normalize pre-synaptic input $I$ such that $I \sim \mathcal{N}(0, u_{th}^2)$, the membrane potential $u$ follows $u \sim \mathcal{N}(\mu_u, \sigma_u^2)$ with $\mu_u = af(u_{th}, u_{rest}, u_c)$ and $\sigma_u^2 = u_{th}^2 h(u_{th}, u_{rest}, u_c, a)$ where $\mu_u$ and $\sigma_u^2$ are directly proportional to the functions $f$ and $h$ respectively. The functions $f$ and $h$ can be approximated as $f(u_{th}, u_{rest}, u_c) = u_{th}^2 + u_{rest}u_c$ and $h(u_{th}, u_{rest}, u_c, a) = 1 + a^2(2u_{th}^2 + (v_c - v_{rest})^2)$.*

The proof of Theorem 1 can be seen in Appendix D and is inspired by Theorem 2 in Zheng et al. (2021). Theorem 1 states that after integrating tdBN normalized pre-synaptic inputs into the QIF neuron according to Equation 11, the membrane potential follows $u \sim \mathcal{N}(\mu_u, \sigma_u^2)$. Therefore, we approximate the values of $\mu_u$ and $\sigma_u^2$ using Theorem 1 to calculate the surrogate gradient window based on the parameters $a, u_{th}, u_{rest}$, and $u_c$, reducing the risk of poor window choice and potential gradient mismatch. We define our new surrogate gradient as

$$\frac{\partial o_n(t)}{\partial u_n(t)} \approx \begin{cases} 1 & \mu_u - \sigma_u \leq u_n(t) \leq \mu_u + \sigma_u \\ 0 & else. \end{cases} \tag{16}$$

To validate Theorem 1 and our new surrogate gradient window in Equation 16, Figure 1 shows our analytical window compared to a static choice of the hyperparameter $\alpha$ across several parameter sets for the QIF neuron model. We compare our window with a common choice for the surrogate gradient window, $\alpha = 1$, as used in Guo et al. (2023); Deng et al. (2023); Duan et al. (2022); Li et al. (2022). In the left histogram, the naïve window almost encompasses the entire distribution, which can lead to a substantial gradient mismatch. Conversely, our analytical window dynamically scales based on the parameter set, fitting the distribution more accurately. The chosen parameter set in the middle figure aligns well with the naïve and analytical windows. However, in the rightmost figure, the naïve window only covers a small portion of the distribution. Since this distribution is not zero-centered, the naïve window additionally fails to account for a significant portion of the spiking activity in the network. Our analytical window addresses this issue by adjusting both the center and width to match the distribution. Therefore, our approach adapts to diverse distribution shapes without requiring detailed knowledge of the underlying voltage distribution or manual window tuning. This minimizes the risk of gradient mismatch and suboptimal surrogate gradient initialization with our QIF neuron model.

## 5 EXPERIMENTS

In this section, we first compare the energy consumption of our QIF neuron model against the standard LIF neuron model across a variety of model architectures and datasets. We then discuss the potential overheads of the QIF neuron model in hardware. Next, we validate the performance of our QIF neuron model using a classification task on static and neuromorphic datasets and compare our results to state-of-the-art works. Finally, we examine the loss landscape, training graphs, and hyperparameter robustness of the QIF and LIF neuron models.

### 5.1 EXPERIMENTAL SETUP

We run our experiments on an Nvidia RTX 3090 GPU and an Intel-12600k CPU with 64 GBs of memory, running Ubuntu 23.04. We use Python 3.12 along with Pytorch 2.4 (Paszke et al., 2019) for the creation and

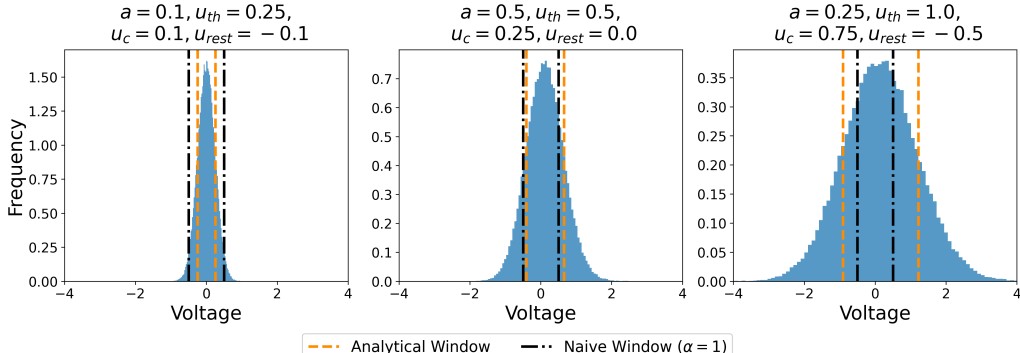

Figure 1: Surrogate gradient window comparison using a naïve and statistical choice of window length with the QIF neuron model using various parameter sets.

training of networks along with loading the CIFAR-10 and CIFAR-100 datasets (Krizhevsky, 2009), Norse 1.1 (Pehle & Pedersen, 2021) as the foundation for our SNN simulations, Tonic 1.4 (Lenz et al., 2021) for loading the N-Cars (Viale et al., 2021) dataset, and SpikingJelly (Fang et al., 2023) for loading CIFAR-10 DVS (Li et al., 2017), N-Caltech-101 (Orchard et al., 2015), and DVS128-Gesture (Amir et al., 2017). We use several model architectures, such as ResNet-19 (Zheng et al., 2021), VGGSNN (Deng et al., 2022), VGG-11 (Kim & Panda, 2021), and DVSGestureNet (Fang et al., 2021), trained on commonly used datasets, such as CIFAR-10/CIFAR-100 (Krizhevsky, 2009), CIFAR-10 DVS (Li et al., 2017), N-Caltech-101 (Orchard et al., 2015), N-Cars (Sironi et al., 2018), and DVS128-Gesture (Amir et al., 2017). We averaged all QIF training results over three training runs using different random number generation seeds and presented the mean $\pm$ standard deviation of our results. Additional details on each dataset, data augmentations, and training setups can be found in Appendices A and B.

## 5.2 SPIKE RATE AND ENERGY CONSUMPTION

To calculate the energy consumption of an SNN, we adopt the same approach as Su et al. (2023), where they approximate it as $E_{SNN} \approx \sum_i E_i$. $E_i$ is the energy consumption of layer $i$ and is defined as

$$E_i = T \cdot (fr \cdot E_{AC} \cdot OP_{AC} + E_{MAC} \cdot OP_{MAC}) \qquad (17)$$

where $T$ is the number of timesteps, $fr$ is the firing rate of layer $i$, $E_{AC}$ and $E_{MAC}$ are the energy consumption of accumulate (AC) and multiply-and-accumulate (MAC) operations respectively, and $OP_{AC}$ and $OP_{MAC}$ are the number of AC and MAC operations of layer $i$. We assume operations take place with 32-bit floating point values on 45nm technology where $E_{MAC} = 4.6pJ$ and $E_{AC} = 0.9pJ$, as done by Su et al. (2023) and other works. We compare the energy consumption of each SNN architecture trained with the QIF and the LIF neuron models with our results showcased in Table 1. To obtain a comparison with the LIF model, we train each model with our implementation of the LIF neuron model. The training setup and hyperparameters for the models trained with the LIF neuron are available in Appendix B. Figure 8, in Ap-

| Neuron Model | CIFAR-10 / ResNet-19 | CIFAR-100 / ResNet-19 | CIFAR-10 DVS / VGGSNN | N-Caltech-101 / VGG-11 | N-Cars / VGGSNN | DVS128-Gesture / DVSGestureNet |
|---|---|---|---|---|---|---|
| LIF | $0.968mJ$ | $0.958mJ$ | $0.848mJ$ | $0.788mJ$ | $1.090mJ$ | $1.095mJ$ |
| QIF | $0.531mJ$ | $0.778mJ$ | $0.361mJ$ | $0.374mJ$ | $0.259mJ$ | $0.724mJ$ |
| **Improvement** | $1.82\times$ | $1.23\times$ | $2.35\times$ | $2.11\times$ | $4.21\times$ | $1.51\times$ |

Table 1: Energy consumption comparison between QIF and LIF neuron models in milliJoules (mJ).

| Neuron Model | CIFAR-10 / ResNet-19 | CIFAR-100 / ResNet-19 | CIFAR-10 DVS / VGGSNN | N-Caltech-101 / VGG-11 | N-Cars / VGGSNN | DVS128-Gesture / DVSGestureNet |
|---|---|---|---|---|---|---|
| LIF | $5.923s$ | $6.356s$ | $2.230s$ | $2.658s$ | $14.999s$ | $1.482s$ |
| QIF | $6.733s$ | $7.602s$ | $2.671s$ | $2.772s$ | $18.030s$ | $2.065s$ |
| **Overhead** | $1.14\times$ | $1.20\times$ | $1.20\times$ | $1.04\times$ | $1.20\times$ | $1.39\times$ |

Table 2: Inference time comparison between QIF and LIF neuron models in seconds (s).

pendix G, showcases the average spike rate of each layer of a ResNet-19 model trained on CIFAR-10 with the QIF and LIF neuron models. On average, our QIF neuron produces around $46\%$ less spiking activity than the LIF neuron. We include similar figures for each of our models and datasets in Appendix G.

Using Equation 17, we calculate the energy consumption of each neuron model in Table 1. We observe energy reduction ranging from $1.23 - 4.21\times$ for the QIF neuron models. These savings are attributed to the non-linear dynamics of the QIF neuron, which tends to induce a voltage distribution with neurons further away from the threshold, as seen in Figure 9. These dynamics increase the difficulty for a neuron to spike, reducing the rate at which less important neurons may fire due to noise or low-quality features. Additionally, neuromorphic datasets show greater energy savings on average than static datasets. This difference may stem from the high sparsity and noise typical of these datasets that cause LIF models to follow noise and produce excess spikes while the QIF models handle this noise more effectively, reducing unnecessary spikes.

To discuss potential concerns related to computational complexity, we showcase the additional latency of a QIF neuron compared to a LIF neuron. A single LIF neuron requires one multiplication and one addition while a single QIF neuron requires three additions and two multiplications. Assuming the two additions required to compute $(u - u_{rest})$ and $(u - u_c)$, from Equation 11, can be done in parallel, the QIF neuron has one addition and multiplication more than the LIF neuron, leading to roughly $2\times$ the computational complexity. On our non-neuromorphic experimental setup, we observe that this leads to inference time overheads between $1.04 - 1.39\times$, as shown in Table 2. Due to the limited public availability of neuromorphic hardware, it is difficult to calculate the exact computational overhead incurred by these additional operations. However, we do know that many neuromorphic hardware implementations, such as Intel Loihi 2 Intel (2021), follow event-driven paradigms. This means the lower spike rate of the QIF neuron has the potential to lower the computational overhead we observed on non-neuromorphic hardware. To put this in perspective, across all datasets and models, the QIF neuron produces an average of $45.47\%$ less spiking activity than the LIF neuron. Therefore, the QIF neuron will only require around half the number of active neurons during inference on average. This suggests that implementing these models on neuromorphic hardware can offset the additional computational complexity of the QIF neuron through its decreased spiking activity.

### 5.3 ACCURACY COMPARISON TO RECENT WORKS

In this section, we compare our QIF model's accuracy to state-of-the-art works that make modifications to the LIF neuron model. Additionally, we include comparisons to recent state-of-the-art results that don't modify the LIF neuron as these techniques could potentially be modified and applied to our QIF model.

As shown in Table 3, our neuron model demonstrates competitive performance on the CIFAR-10 dataset, matching the performance of other neuron model optimizations within $1\%$ accuracy on average, such as those presented in Lian et al. (2024; 2023); Yao et al. (2022); Fang et al. (2021) with 2 timesteps and being slightly outperformed by these works with 4 timesteps. When compared with alternative approaches, our method surpasses most others, though we observe approximately a $2\%$ decrease in accuracy relative to the top-performing methods in Mukhoty et al. (2023); Guo et al. (2023); Deng et al. (2023). On CIFAR-100, our model matches or outperforms other neuron model works at 2 timesteps and is marginally outperformed by Lian et al. (2024) and Yao et al. (2022) with 4 timesteps. Compared to dissimilar techniques, only the

| Work | Method | Timesteps | CIFAR-10 Accuracy | CIFAR-100 Accuracy |
|------|--------|-----------|-------------------|--------------------|
| STDP-tdBN Zheng et al. (2021) | Batch Normalization | 6 | 93.16% | 71.12% |
| TEBN Duan et al. (2022) | Batch Normalization | 6 | 94.71% | 76.41% |
| MPBN Guo et al. (2023) | Membrane Normalization | 2 | 96.47% | 79.51% |
| TET Deng et al. (2022) | Loss Function | 6 | 94.50% | 74.72% |
| Surrogate Module Deng et al. (2023) | Hybrid | 4 | 96.82% | 79.18% |
| LocalZO + TET Mukhoty et al. (2023) | Direct Training | 2 | 95.03% | 76.36% |
| Dspike Li et al. (2021b) | Surrogate Gradient | 2 | 93.13% | 71.68% |
| IM-Loss Guo et al. (2022) | Loss Function + SG | 2 | 93.85% | 70.18% |
| GLIF Yao et al. (2022) | Neuron Model | 4 | 94.85% | 77.05% |
|  |  | 2 | 94.44% | 75.48% |
| LSG Lian et al. (2023) | Neuron Model + SG | 4 | 95.17% | 76.85% |
|  |  | 2 | 94.41% | 76.32% |
| IM-LIF Lian et al. (2024) | Neuron Model | 3 | 95.29% | 77.21% |
| **QIF (Ours)** | **Neuron Model** | **4** | **94.52 ± 0.12%** | **76.89 ± 0.17%** |
|  |  | **2** | **94.44 ± 0.07%** | **76.80 ± 0.06%** |

Table 3: Summary and comparison of results on static datasets. Acronyms: Surrogate Gradient (SG).

work of Deng et al. (2023) and Guo et al. (2023) showed significantly better results. All compared works, including ours, use the ResNet-19 architecture on CIFAR-10 and CIFAR-100.

Next, we look at the training results on neuromorphic datasets in Table 4. On CIFAR-10 DVS, we exceed the accuracy of all other neuron model approaches on average by 7%. Even when compared to dissimilar methods, we outperform the best-performing approach from Deng et al. (2023) by over 3% and surpass all other methods by more than 8%. For the N-Caltech-101 dataset, our model achieves the highest accuracy, outperforming the LIF neuron model work of Li et al. (2022) under identical conditions by nearly 2%. Similarly, on N-Cars, we see a 3% or greater accuracy boost over the LIF neuron, without requiring data augmentations. Lastly, on the DVS128-Gesture dataset, we fall short of Fang et al. (2021) and Lian et al. (2024) by 1% accuracy. However, we only use half and a quarter of the timesteps as these works, respectively. Still, we outperform most works utilizing other methods, with only Mukhoty et al. (2023) outperforming the QIF model by just over 1% accuracy.

These results showcase the QIF model's ability to match or outperform the LIF model on a variety of neuromorphic and static datasets. Although works employing dissimilar techniques demonstrate superior performance on specific datasets, exploring how these methods can be adapted and integrated with the QIF neuron model to enhance performance remains an interesting path. We include additional experiments with a larger ResNet model and vision transformer architectures in Appendix E.

## 5.4 Loss Landscapes

To evaluate the training improvements of deep spiking neural networks using the QIF neuron model, we analyze the loss landscape of identical model architectures trained with both QIF and LIF neurons. The loss landscapes are visualized using the method described in Li et al. (2018). As shown in Figure 15, the loss landscape for a model trained with QIF neurons is significantly broader compared to a model trained with LIF neurons. This broader landscape includes a wider local minimum and smoother surface which can facilitate faster convergence and improved performance, as seen in Figure 17. In contrast, the narrower loss landscape of the LIF model necessitated reducing the initial learning rate when training on the CIFAR-10 and CIFAR-100 datasets. As discussed in Section 5.2, the non-linear dynamics of the QIF neuron introduce greater spiking difficulty, which allows QIF models to focus on learning the most relevant features rather than noise, contributing to its faster convergence relative to LIF models.

| Dataset | Work | Method | Architecture | Timesteps | Accuracy |
|---|---|---|---|---|---|
| CIFAR-10 DVS | TEBN Duan et al. (2022) | Batch Normalization | 7-Layer CNN | 10 | 75.10% |
| | MPBN Guo et al. (2023) | Membrane Normalization | ResNet-20 | 10 | 78.70% |
| | Dspike Li et al. (2021b) | Surrogate Gradient | ResNet-18 | 10 | 75.40% |
| | TET Deng et al. (2022) | Loss Function | VGGSNN | 10 | 77.40% |
| | IM-Loss Guo et al. (2022) | Loss Function + SG | ResNet-19 | 10 | 72.60% |
| | Surrogate Module Deng et al. (2023) | Hybrid | ResNet-18 | 10 | 83.19% |
| | LocalZO + TET Mukhoty et al. (2023) | Direct Training | VGGSNN | 10 | 75.62% |
| | LIF w/ NDA Li et al. (2022) | Data Augmentations | VGG-11 | 10 | 79.60% |
| | PLIF Fang et al. (2021) | Neuron Model | 7-Layer CNN | 20 | 74.80% |
| | GLIF Yao et al. (2022) | Neuron Model | ResNet-19 | 16 | 78.10% |
| | LSG Lian et al. (2023) | Neuron Model + SG | VGGSNN | 10 | 77.90% |
| | IM-LIF Lian et al. (2023) | Neuron Model | VGGSNN | 10 | 80.50% |
| | **QIF (Ours)** | **Neuron Model** | **VGGSNN** | **10** | **86.80 ± 1.12%** |
| N-Caltech-101 | HATS Sironi et al. (2018) | Histogram | SVM | × | 64.20% |
| | DART Ramesh et al. (2020) | Histogram | SVM | × | 66.80% |
| | SALT Kim & Panda (2021) | BN + SALT | VGG-11 | 20 | 55.00% |
| | LocalZO + TET Mukhoty et al. (2023) | Direct Training | VGGSNN | 10 | 79.86% |
| | LIF w/ NDA Li et al. (2022) | Data Augmentations | VGG-11 | 10 | 78.20% |
| | **QIF w/ NDA (Ours)** | **Neuron Model** | **VGG-11** | **10** | **80.01 ± 0.05%** |
| N-Cars | HATS Sironi et al. (2018) | Histogram | SVM | × | 81.00% |
| | CarSNN Viale et al. (2021) | Direct Training | 4-Layer CNN | 10 | 77.0% |
| | LocalZO + TET Mukhoty et al. (2023) | Direct Training | VGGSNN | 10 | 96.78% |
| | LIF w/ NDA Li et al. (2022) | Data Augmentations | VGG-11 | 10 | 90.10% |
| | **QIF (Ours)** | **Neuron Model** | **VGGSNN** | **10** | **93.68 ± 0.15%** |
| DVS128-Gesture | RSNN Xu et al. (2024) | Recurrent SNN | 4-Layer RSNN | 20 | 95.80% |
| | DECOLLE Kaiser et al. (2020) | Online Learning | 6-Layer SCNN | 500 | 95.54% |
| | SLAYER Shrestha & Orchard (2018) | Direct Training | 8-Layer SCNN | 5 | 93.64% |
| | LocalZO + TET Mukhoty et al. (2023) | Direct Training | VGGSNN | 10 | 98.04% |
| | PLIF Fang et al. (2021) | Neuron Model | DVSGestureNet | 20 | 97.57% |
| | IM-LIF Lian et al. (2023) | Neuron Model | VGGSNN | 40 | 97.33% |
| | **QIF (Ours)** | **Neuron Model** | **DVSGestureNet** | **10** | **96.76 ± 0.43%** |

Table 4: Comparison between state-of-the-art techniques and the QIF neuron model on neuromorphic datasets. Acronyms: Spiking Convolutional Neural Network (SCNN), Recurrent SNN (RSNN), Neuromorphic Data Augmentations (NDA), Surrogate Gradient (SG).

Additional visualizations of loss contours, loss surfaces, and training graphs for all models and datasets are provided in Appendix H. Furthermore, we include a robustness study for the QIF and LIF neurons to their hyperparameters in Appendix F.

## 6 CONCLUSION

In this work, we introduced a discretized Quadratic Integrate-and-Fire (QIF) neuron model to address the limitations of the LIF neuron models' linear voltage dependence. We provide an analytical method for calculating surrogate gradient windows enables efficient training of these networks, reducing the risk of gradient mismatch and improving training stability. Additionally, we showcased substantial energy savings when comparing model architectures using the QIF and LIF neuron models and discussed how neuromorphic hardware can reduce the computational overhead of the QIF neuron model. Our evaluation also demonstrates that the QIF model not only performs competitively on static datasets but can also achieve significant accuracy improvements on neuromorphic datasets. Overall, our results show that the QIF neuron model offers a promising direction for energy efficiency and performance in deep-spiking neural networks, particularly when deployed on neuromorphic hardware.

# 7    REPRODUCIBILITY STATEMENT

To recreate our results, one can look at the following information. Appendix A details each dataset and augmentation applied, and Appendix B provides references to model architectures along with hyperparameters used for training. Additionally, an anonymous repository of our project has been included with this submission, providing details on recreating and running our experiments. The exact system setup and core dependencies are detailed in Section 5.1 with versioning of other dependencies detailed in the included repository. This repository also details the steps required to recreate our figures, such as the ones found in Appendices G and H. Finally, we have included the discretization steps for the QIF neuron in Appendix C and proofs of novel claims in Appendix D.

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

# A DATASETS AND AUGMENTATIONS

## A.1 CIFAR-10

**CIFAR-10** (Krizhevsky, 2009) is a widely used dataset for traditional ANN and SNN models. It consists of $60,000$, $32 \times 32$ colored images consisting of 10 classes, with some examples being airplanes, automobiles, cats, and horses. There are $6,000$ images per class. Additionally, the dataset is split into standardized training and testing sets with $50,000$ and $10,000$ images, respectively. When training, we perform the following dataset augmentations: Random cropping after adding 4 pixels of zero padding to the outside of the image, random horizontal flipping, cutout, and normalization of each image by the mean and standard deviation of the dataset.

## A.2 CIFAR-100

**CIFAR-100** (Krizhevsky, 2009) is also a widely used dataset for traditional ANN and SNN models. It contains the same shape and number of images as CIFAR-10. However, CIFAR-100 contains 100 unique classes instead of 10, with several examples being bed, rocket, apples, and otter. There are only 600 images per class, with each class being evenly split between standardized training and testing sets the same size as CIFAR-10. We use the same augmentations as with CIFAR-10 in addition to AutoAugment (Cubuk et al., 2019).

## A.3 CIFAR-10 DVS

**CIFAR-10 DVS** (Li et al., 2017) uses a subset of CIFAR-10 with $1,000$ images from each class. To create this dataset, the authors first place images from their subset on a large LCD monitor. Then, they aim a Digitial Vision Sensor (DVS) at the LCD monitor and perform a pan and tilt to generate spiking events with size $128 \times 128$ with two polarity channels. These events can then be accumulated for a set number of timesteps to generate frames of spiking activity. When using this dataset, we accumulate events into 10 frames and resize them to $48 \times 48$. Additionally, we apply random horizontal flipping and randomly rotate the image up to $\pm 10°$.

## A.4 N-CALTECH-101

**N-Caltech-101** (Orchard et al., 2015) was created similarly to CIFAR-10 DVS. First, the authors select 8831 images from the original Caltech-101 dataset, removing the 'Faces' class due to conflicts with the 'Faces Easy' class. Next, the authors use a DVS, similar to the CIFAR-10 DVS dataset, to transform the dataset into spiking events. Similarly to CIFAR-10 DVS, we accumulate events into 10 frames and resize them to $48 \times 48$. We then apply the $M1N1$ neuromorphic data augmentation policy described by (Li et al., 2022).

## A.5 N-CARS

**N-Cars** (Sironi et al., 2018) contains two classes, *Car* and *Background*, with $12,336$ and $11,693$ samples respectively being the same size and shape of CIFAR-10 DVS samples. The dataset was generated by attaching a DVS camera to the windshield of a car and driving around in various sessions. The dataset contains a standard training and testing set with $15,422$ and $8,607$ samples, respectively. The only preprocessing we perform on this dataset is rescaling images to be $48 \times 48$ after accumulating events into 10 frames.

## A.6 DVS128-GESTURE

**DVS128-Gesture** (Amir et al., 2017) contains $1,342$ samples of various gestures, such as waving, being performed by 29 different individuals in front of a DVS camera under 3 different lighting conditions. These gestures are recorded with the same size and shape as CIFAR-10 DVS samples. After accumulating events into 10 frames, we randomly roll the pixels of the frames by up to 5 pixels in either the $x$ or $y$ axis.

## B TRAINING SETUP

We use the following model architectures: ResNet-19 Zheng et al. (2021), VGGSNN Deng et al. (2022), VGG-11 Kim & Panda (2021), and DVSGestureNet Fang et al. (2021). Complete details of the training setup and hyperparameters used for each dataset and model can be seen in Table 5. We use a cosine decay learning rate scheduler to slowly decay the learning rate to 0 for all models. When using stochastic gradient descent (SGD), we use a momentum of $0.9$. When using Adam, we set $\beta_1 = 0.9$, $\beta_2 = 0.999$. For all models trained with the QIF neuron model, we use the following parameters: $u_{th} = 0.5$, $u_c = 0.5$, $u_{rest} = 0$, and $a = 0.25$. When training with the LIF neuron model, we adopt the same model and training setup in Table 5, except we use a learning rate of $1e - 3$ when training with the ResNet-19 model. We set the LIF neuron parameters as $u_{th} = 0.5$, $u_{rest} = 0$, $\beta = 0.25$, and we use the surrogate gradient defined in Equation 5, with $\alpha = 1$.

| Parameters | CIFAR-10 | CIFAR-100 | CIFAR-10 DVS | N-Caltech-101 | N-Cars | DVS128-Gesture |
|---|---|---|---|---|---|---|
| Model | ResNet-19 | ResNet-19 | VGGSNN | VGG-11 | VGGSNN | DVSGestureNet |
| Optimizer | SGD | SGD | Adam | Adam | Adam | Adam |
| Weight Decay | $1e-4$ | $1e-4$ | $5e-4$ | $1e-4$ | $1e-4$ | $1e-4$ |
| Learning Rate | 0.1 | 0.1 | $1e-3$ | $1e-3$ | $1e-3$ | $1e-3$ |
| Epochs | 350 | 350 | 100 | 100 | 100 | 100 |
| Batch Size | 128 | 128 | 64 | 64 | 64 | 32 |
| Timesteps | 2 | 2 | 10 | 10 | 10 | 10 |
| Dropout | $\times$ | $\times$ | 0.6 | 0.6 | 0.6 | 0.75 |

Table 5: Training setup for each dataset

## C QUADRATIC INTEGRATE-AND-FIRE NEURON MODEL DISCRETIZATION

The QIF neuron model is defined as

$$\tau \frac{du}{dt} = a(u - u_{rest})(u - u_c) + RI, \tag{18}$$

where $\tau$ is a membrane time constant, $u$ is the neurons voltage, $a$ is a sharpness parameter, $u_{rest}$ is the resting voltage, $u_c$ is the critical spiking threshold, $R$ is a resistor, and $I$ is pre-synaptic input. We use Euler's method to discretize Equation 18, as done by Wu et al. (2018). First, we replace the derivative with the following approximation

$$\frac{du}{dt} \approx \frac{u(t+1) - u(t)}{\Delta t}. \tag{19}$$

Substituting this into Equation 18, we have

$$\tau \frac{u(t+1) - u(t)}{\Delta t} \approx a(u(t) - u_{rest})(u(t) - u_c) + RI(t), \tag{20}$$

with $t$ being some discrete timestep and $\Delta t$ being some small step size. Then, solving for $u(t+1)$ gives us

$$u(t+1) \approx u(t) + \frac{\Delta t}{\tau}[a(u(t) - u_{rest})(u(t) - u_c) + RI(t)]. \tag{21}$$

Next, assuming that $\Delta t = 1$, $\frac{R}{\tau} = 1$, and $\frac{1}{\tau}$ has been folded into $a$, we can simplify our equation to

$$u(t+1) \approx u(t) + a(u(t) - u_{rest})(u(t) - u_c) + I(t). \tag{22}$$

When Equation 18 has zero input, i.e. $I(t) = 0$ for all $t$, its $u_{rest}$ and $u_c$ are the roots of the polynomial (Gerstner et al., 2014). Therefore, to ensure our discretized model satisfies this behavior, we drop the additional $u(t)$ term in Equation 22 and obtain our final discretization defined as

$$u(t+1) \approx a(u(t) - u_{rest})(u(t) - u_c) + I(t). \tag{23}$$

We believe that dropping the $u(t)$ term allows for better interpretability of the dynamics invoked by different parameter choices for the QIF neuron. We performed preliminary testing with the additional $u(t)$ where we observed higher spiking activity with no noticeable performance improvement.

## D PROOFS OF THEOREMS

**Theorem 1.** *Under the discrete QIF neuron model using tdBN to normalize pre-synaptic input $I$ such that $I \sim \mathcal{N}(0, u_{th}^2)$, the membrane potential $u$ follows $u \sim \mathcal{N}(\mu_u, \sigma_u^2)$ with $\mu_u = af(u_{th}, u_{rest}, u_c)$ and $\sigma_u^2 = u_{th}^2 h(u_{th}, u_{rest}, u_c, a)$ where $\mu_u$ and $\sigma_u^2$ are directly proportional to the functions $f$ and $h$ respectively. The functions $f$ and $h$ can be approximated as $f(u_{th}, u_{rest}, u_c) = u_{th}^2 + u_{rest}u_c$ and $h(u_{th}, u_{rest}, u_c, a) = 1 + a^2(2u_{th}^2 + (v_c - v_{rest})^2)$.*

*Proof.* We define the discrete QIF neuron model as

$$u(t+1) = a(u(t) - u_{rest})(u(t) - u_c) + I(t), \tag{24}$$

where $t$ is the timestep, $u$ is the membrane potential, $a$ is a sharpness parameter, $u_{rest}$ is the resting voltage, $u_c$ is the critical firing voltage, and $I$ is pre-synaptic input. Considering the membrane potential $u(t)$ and assuming the last firing time was $t' < t$, we have

$$u(t+1) \approx \sum_{k=t'+1}^{t} a^{t-k-1}(I(k-1) - u_{rest})(I(k-1) - u_c) + I(k). \tag{25}$$

This approximation only holds if $a$ is a relatively small constant. In our work, $a$ is typically set to $0.25$. Small values of $a$ ensure that input into the neuron more than two timesteps ago has a minuscule impact on the voltage at timestep $t+1$, meaning we can simply Equation 25 as

$$u(t+1) \approx a(I(t-1) - u_{rest})(I(t-1) - u_c) + I(t). \tag{26}$$

Then, under the tdBN assumption that $I \sim \mathcal{N}(0, u_{th}^2)$ and assuming that $I(t)$ is an independent and identically distribution sample (i.i.d) for all $t$, we can approximate the expectation of $u(t+1)$ as

$$\begin{aligned}
\mathbb{E}[u(t+1)] &\approx \mathbb{E}[a(I(t-1) - u_{rest})(I(t-1) - u_c) + I(t)] \\
&= \mathbb{E}[a(I(t-1)^2 - I(t-1)(u_{rest} + u_c) + u_{rest}u_c) + I(t)] \\
&= \mathbb{E}[aI(t-1)^2] - \mathbb{E}[aI(t-1)(u_{rest} + u_c)] + \mathbb{E}[au_{rest}u_c] + \mathbb{E}[I(t)] \quad \text{(i.i.d)} \\
&= a\mathbb{E}[I(t-1)^2] - a(u_{rest} + u_c)\mathbb{E}[I(t-1)] + au_{rest}u_c \\
&= a(u_{th}^2 + u_{rest}u_c).
\end{aligned}$$

Likewise, we can approximate the variance of $u(t+1)$ as

$$
\begin{aligned}
Var[u(t+1)] &\approx Var[a(I(t-1) - u_{rest})(I(t-1) - u_c) + I(t)] \\
&= Var[a(I(t-1)^2 - I(t-1)(u_{rest} + u_c) + u_{rest}u_c) + I(t)] \\
&= Var[aI(t-1)^2] + Var[aI(t-1)(u_{rest} + u_c)] + Var[au_{rest}u_c] + Var[I(t)] \quad \text{(i.i.d)} \\
&= a^2 Var[I(t-1)^2] + a^2(u_{rest} + u_c)^2 Var[I(t-1)] + Var[I(t)] \\
&= u_{th}^2(1 + a^2(2u_{th}^2 + (v_c + v_{rest})^2)).
\end{aligned}
$$

Therefore, we can define functions $f : \mathbb{R}^3 \to \mathbb{R}$ and $h : \mathbb{R}^4 \to \mathbb{R}$ as

$$f(u_{th}, u_{rest}, u_c) = u_{th}^2 + u_{rest}u_c$$
$$h(u_{th}, u_{rest}, u_c, a) = 1 + a^2(2u_{th}^2 + (v_c + v_{rest})^2).$$

Then $\mu_u \approx af(u_{th}, u_{rest}, u_c)$ and $\sigma_u^2 \approx u_{th}^2 h(u_{th}, u_{rest}, u_c, a)$, thus showing that $u \sim \mathcal{N}(\mu_u, \sigma_u^2)$.  □

## E  ADDITIONAL EXPERIMENTS

### E.1  RESNET-34 ON CIFAR-10

To showcase the QIF neuron model's ability to scale to larger and deeper model architectures, we train a ResNet-34 Zheng et al. (2021) on CIFAR-10 with the images scaled up to $64 \times 64$. We choose ResNet-34 as it has around $2\times$ the parameters as the ResNet-19 architecture. We use mostly the same hyperparameters and dataset augmentations as we did when using ResNet-19, only changing two parameters. The surrogate gradient window of the LIF model is set to $\alpha = 0.5$ and the learning rate for the LIF model is set to $0.01$.

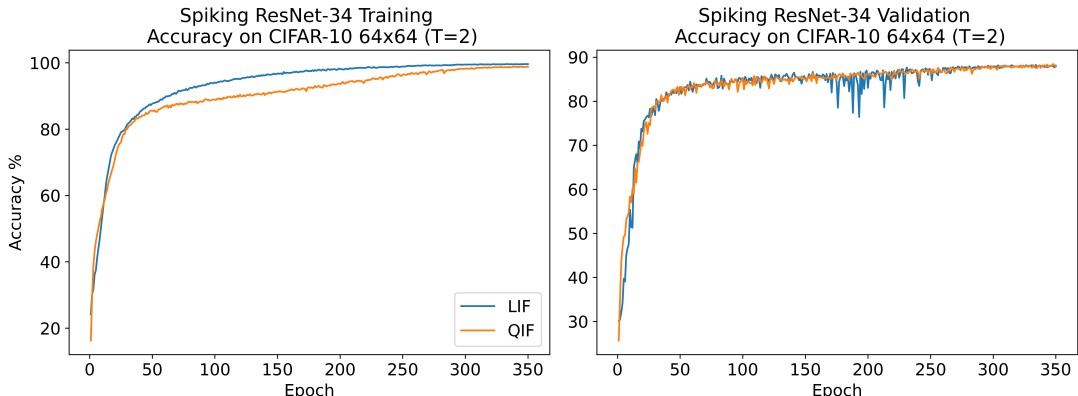

Figure 2: Training and validation accuracy comparison of ResNet-34 on CIFAR-10 64x64 validation using QIF and LIF neuron models.

Figure 2 showcases the training of the QIF and LIF neuron models. In this figure, we see that the LIF neuron model starts to outperform the QIF neuron model in terms of training accuracy at around 50 epochs into training. However, we see much more volatility in the LIF neuron model validation accuracy, leading us to believe the LIF model is overfitting. On the other hand, the QIF neuron model has a much smoother and less volatile validation accuracy throughout the training process. Both models obtain similar validation accuracies at $88.24\%$ for the LIF model and $88.50\%$ for the QIF model.

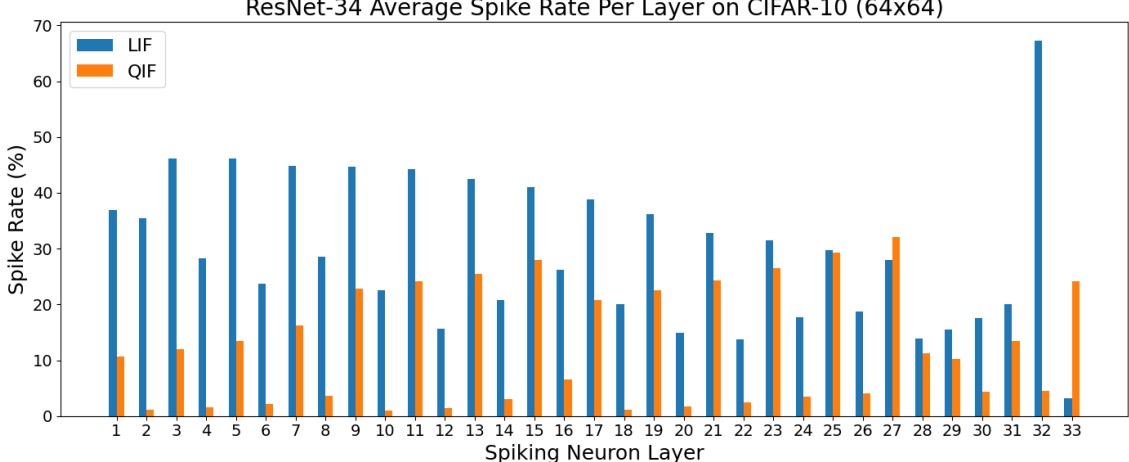

Figure 3: Average spike rate comparison of ResNet-19 over the CIFAR-10 validation set using QIF and LIF neuron models.

Figure 3 showcases the average spike rate per layer of the QIF and LIF neuron models where we see significant reductions in spiking activity from the QIF neuron model. Using these spike rates to calculate the energy consumption of both models we get that the LIF model consumes approximately $3mJ$ while the QIF model consumes approximately $2.15\times$ less energy at $1.4mJ$. These results showcase the QIF neuron model's ability to scale to larger CNN architectures while providing competitive performance and maintaining high energy savings.

### E.2 META-SPIKEFORMER ON TINY IMAGENET

To showcase the QIF neuron models' performance on a non-convolutional architecture, we perform a preliminary training experiment using the 31.3 million parameter Meta-SpikeFormer architecture Yao et al. (2024) on the Tiny ImageNet dataset Deng et al. (2009). We use the same hyperparameters as noted in the work of Yao et al. (2024) for both neuron models. We use the same dataset augmentations that we applied to CIFAR-10. We use the same neuron parameters as we did for ResNet-19, except we change the surrogate gradient window for the LIF neuron model to $\alpha = 0.5$. Additionally, we replace all batch normalizations with Threshold-Dependent Batch Normalizations (tdBN) Zheng et al. (2021).

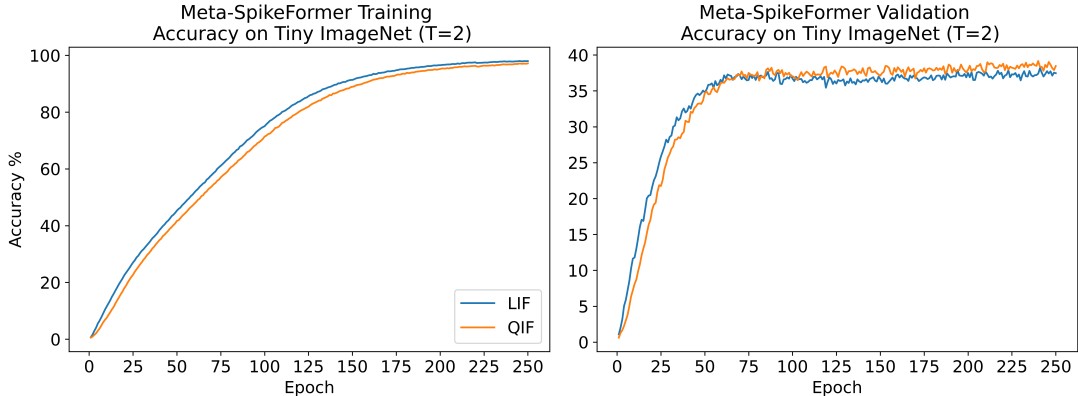

Figure 4: Training and validation accuracy comparison of ResNet-34 on CIFAR-10 64x64 validation using QIF and LIF neuron models.

The training results in Figure 4 show that the LIF model maintains a slight lead in training accuracy throughout training. When looking at validation accuracy, we see the LIF model outperforms the QIF model up until around epoch 65, where both neuron models have similar accuracy. At around epoch 80, the QIF neuron starts to pull away, consistently having higher validation accuracy for the rest of the training. The QIF and LIF models achieve validation accuracies of 39.16% and 38.16%, respectively.

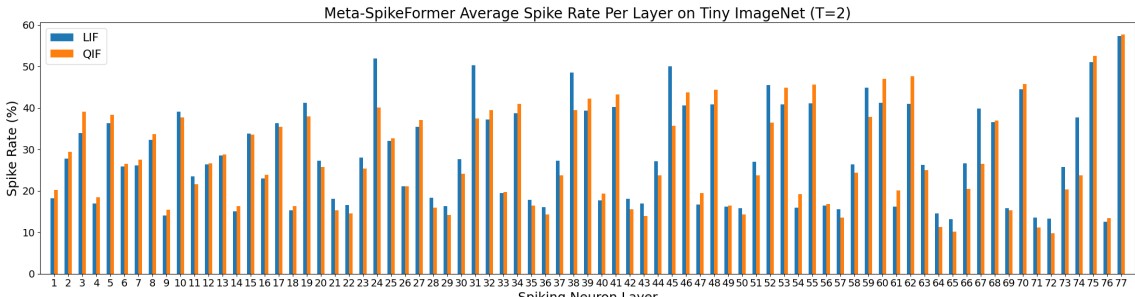

Figure 5: Average spike rate comparison of ResNet-19 over the CIFAR-10 validation set using QIF and LIF neuron models.

Figure 5 showcases the spiking activity of both neuron models in each layer of the network. On average, the QIF neuron model spikes $4\%$ less than the LIF neuron model. When calculating the energy consumption difference, the QIF model consumes approximately $8.20mJ$ while the LIF model consumes approximately $0.99\times$ less energy at $8.14mJ$. Due to the marginal difference in spike rate, energy consumption, and validation accuracy, the LIF neuron model may be preferred for this task due to its reduced computational complexity.

As a side note, tdBN is required for the analytical calculation of the QIF neuron model's surrogate gradient window, so we modified the architecture accordingly. While tdBN has been extensively tested on convolutional neural network architectures, existing spiking vision transformer works do not utilize tdBN techniques

to the best of our knowledge. Therefore, we are unsure of how this decision affected model performance and spike rate. Examining whether tdBN is appropriate for usage within spiking vision transformer architectures remains an interesting future research direction.

## F ROBUSTNESS ANALYSIS

In this section, we examine the robustness of the QIF and LIF neuron models in relation to hyperparameter choice. To do this, we use the LeNet-5 (Lecun et al., 1998) architecture trained on the Fashion-MNIST (F-MNIST) dataset (Xiao et al., 2017) and DVSGestureNet Fang et al. (2021) architecture trained on DVS128-Gesture (Amir et al., 2017). We alter the LeNet-5 architecture in two ways. We perform tdBN after each convolutional layer, and we alter the classifier to now only contain two layers with 120 and 10 hidden units, respectively. Additionally, for the LeNet-5 architecture, each model was trained for 10 epochs using the Adam optimizer with a learning rate of $1e - 3$, weight decay of $1e - 4$, batch size of 128, 2 timesteps, and the same random number generation seed for all models. For the DVSGestureNet architecture, we follow the same training setup noted in Appendix B with two modifications. We reduce the training time to just 20 epochs and remove the dropout layers. The LIF neuron used $\alpha = 1$ for its surrogate gradient for both models while the QIF model used our analytical equation. For LIF neurons, we sweep through the threshold, $u_{th}$, and decay, $\beta$, hyperparameters while for QIF neurons, we sweep through the threshold, $u_{th}$, critical voltage threshold, $u_c$, and sharpness parameter, $a$. For both models, we keep the resting voltage at a constant zero. We perform a grid search over all hyperparameters listed above with the values $\{0.2, 0.4, 0.6, 0.8, 1.0\}$

Figure 6 and 7 showcase the results of our hyperparameter sweep. While these figures showcase the results of our hyperparameter sweep, they include parameter combinations that don't make sense under the assumptions in the proof of Theorem 1. Specifically, when using tdBN, we assume that $a$ is a relatively small constant. Therefore, parameter sets with $a > 0.4$ are unrealistic choices. However, we include values of $a > 0.4$ in our figures to showcase the performance of the QIF neuron model with naïve parameter choices. We report the mean $\pm$ standard deviation, minimum, and maximum of each neuron model's accuracy values in Table 6. When calculating the values in Table 6, we exclude results from parameter sets with $a > 0.4$. These figures and the table showcase that under reasonable parameter selection, the QIF model outperforms the LIF in terms of minimum, average, and maximum accuracies while having a smaller standard deviation. In the case of LeNet-5, the results are relatively close with only minor differences. However, with DVSGestureNet, we see the QIF model greatly outperforming the LIF model in all metrics. These results indicate the QIF neuron model matches or surpasses the LIF neuron model in terms of hyperparameter robustness.

| Model | Mean Accuracy | Minimum Accuracy | Maximum Accuracy |
|---|---|---|---|
| LIF LeNet-5 | $88.32 \pm 0.41\%$ | $87.26\%$ | $88.99\%$ |
| QIF LeNet-5 ($a \leq 0.4$) | $88.73 \pm 0.25\%$ | $88.00\%$ | $89.08\%$ |
| LIF DVSGestureNet | $79.08 \pm 4.91\%$ | $69.44\%$ | $86.81\%$ |
| QIF DVSGestureNet ($a \leq 0.4$) | $91.33 \pm 2.38\%$ | $84.72\%$ | $95.14\%$ |

Table 6: Mean, minimum, and maximum accuracies obtained from hyperparameter sweep using LeNet-5 trained on Fashion-MNIST and DVSGestureNet trained on DVS128-Gesture with the QIF and LIF neuron models.

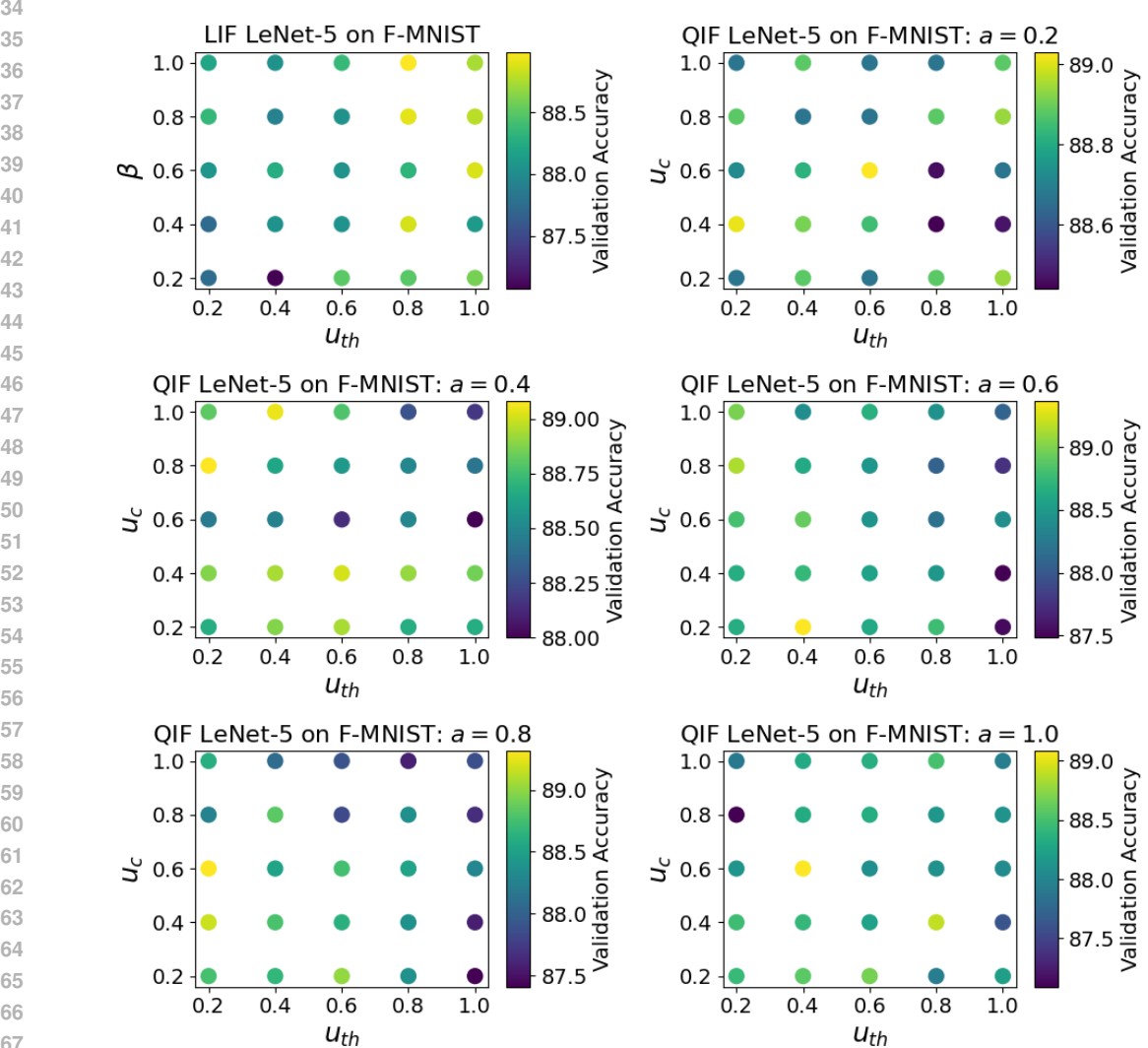

Figure 6: Hyperparameter sweep for LeNet-5 trained with both the QIF and LIF neuron models on Fashion-MNIST.

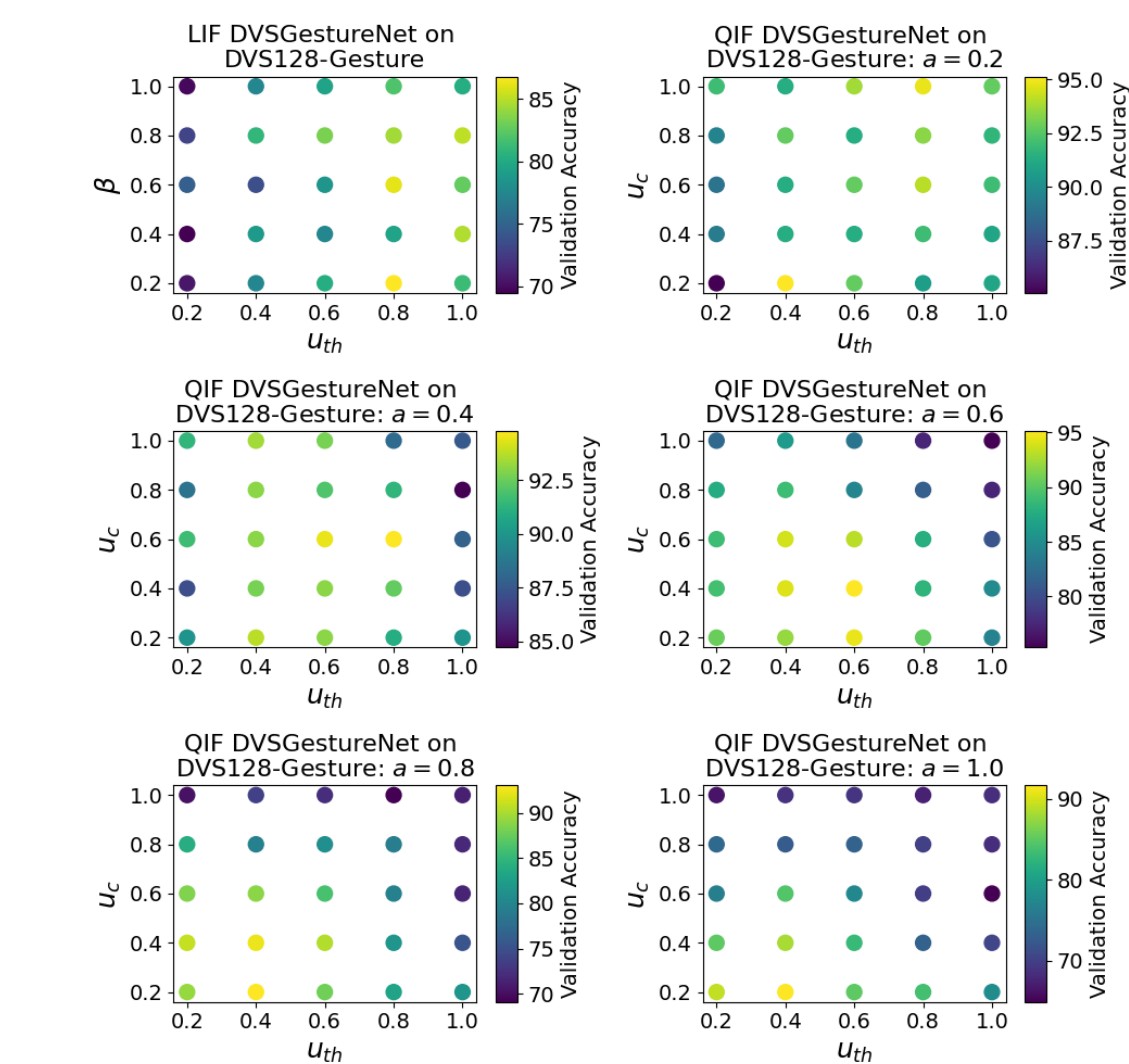

Figure 7: Hyperparameter sweep for DVSGestureNet trained with both the QIF and LIF neuron models on DVS128-Gesture.

## G   Spike Rates of SNN Models

In this section, we showcase the average spike rate per layer of each model and dataset averaged over the entire validation set. We train the same model architecture twice using the LIF and QIF neurons. Across all models, the average spike rate across all layers is lower for the QIF neuron model. We also provide greater introspection into the spike rate by analyzing the voltage distributions of the LIF and QIF neuron models.

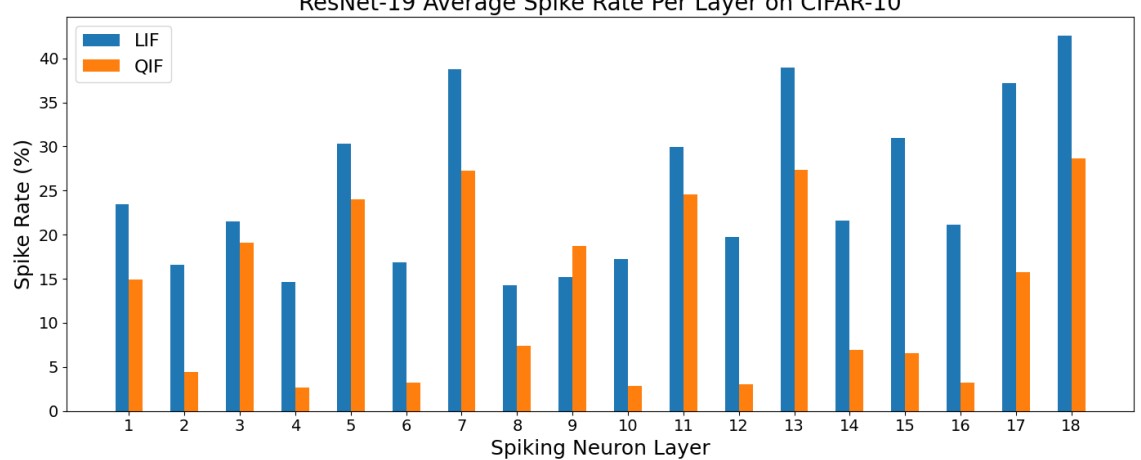

Figure 8: Average spike rate comparison of ResNet-19 over the CIFAR-10 validation set using QIF and LIF neuron models.

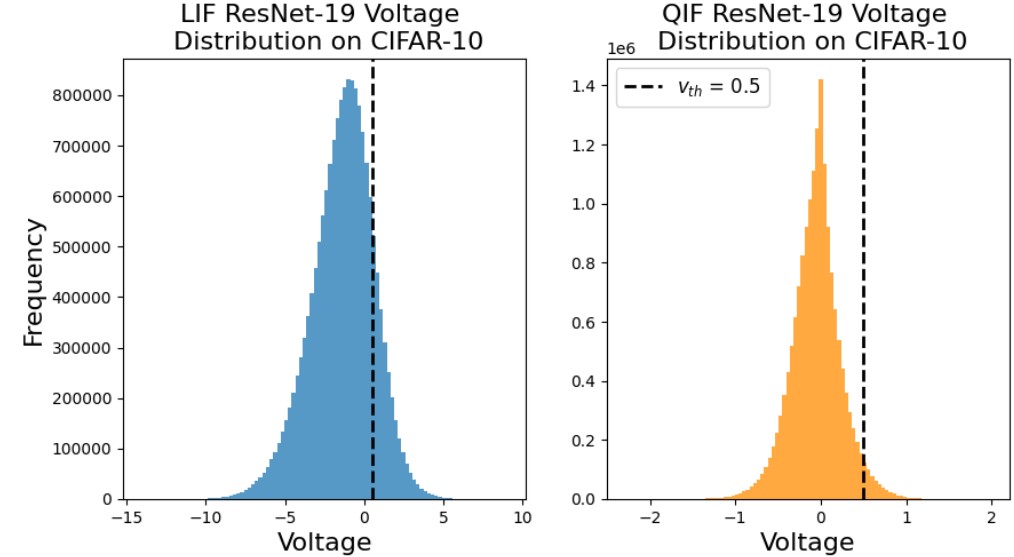

Figure 9: Voltage distribution comparison of the LIF and QIF neuron models using ResNet-19 trained on CIFAR-10. The distributions are taken from the 6th layer of neurons in the network when inferencing across the entire validation set. On the left, the LIF neuron model creates a broad distribution, with around $16\%$ of all neurons being greater than the threshold ($v_{th} = 0.5$). On the right, the QIF neuron model creates a much narrower distribution with tight grouping around zero. This leads to only around $4\%$ of the neurons being above the threshold ($v_{th} = 0.5$).

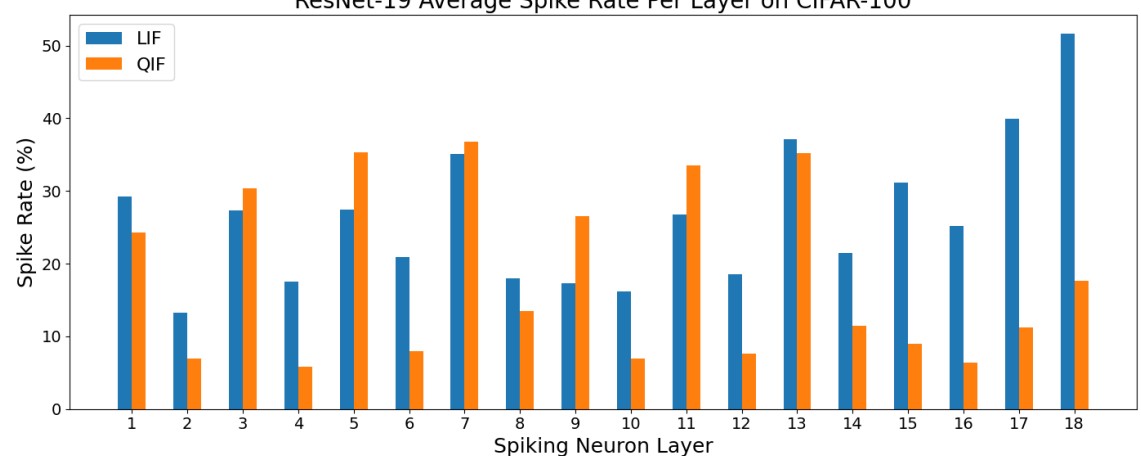

Figure 10: Average spike rate comparison of ResNet-19 over the CIFAR-100 validation set using QIF and LIF neuron models.

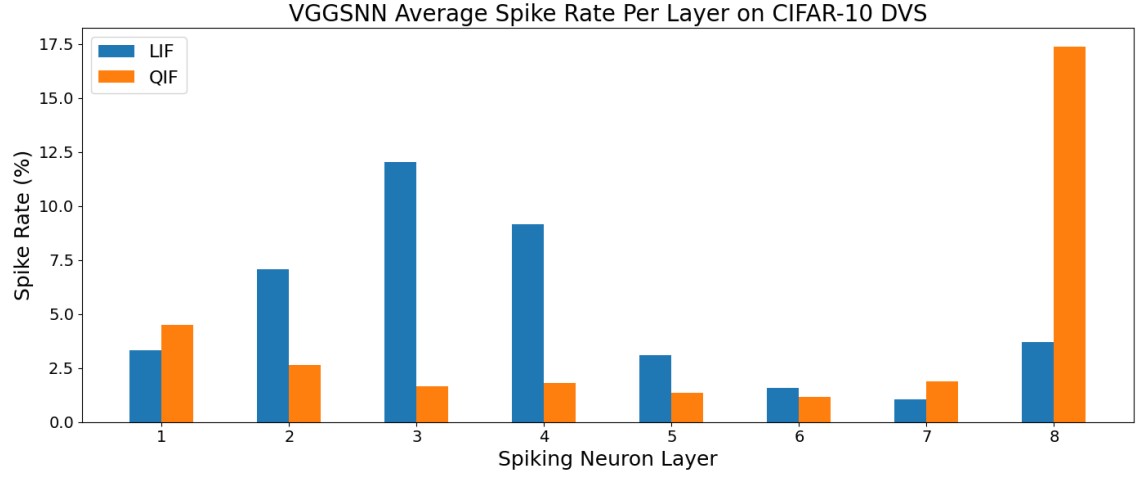

Figure 11: Average spike rate comparison of VGGSNN over the CIFAR-10 DVS validation set using QIF and LIF neuron models.

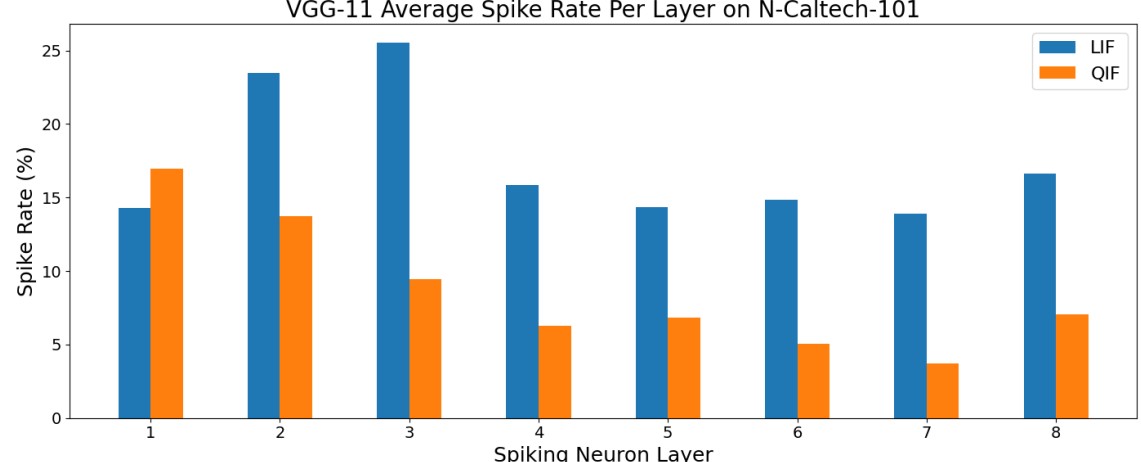

Figure 12: Average spike rate comparison of VGG-11 over the N-Caltech-101 validation set using QIF and LIF neuron models.

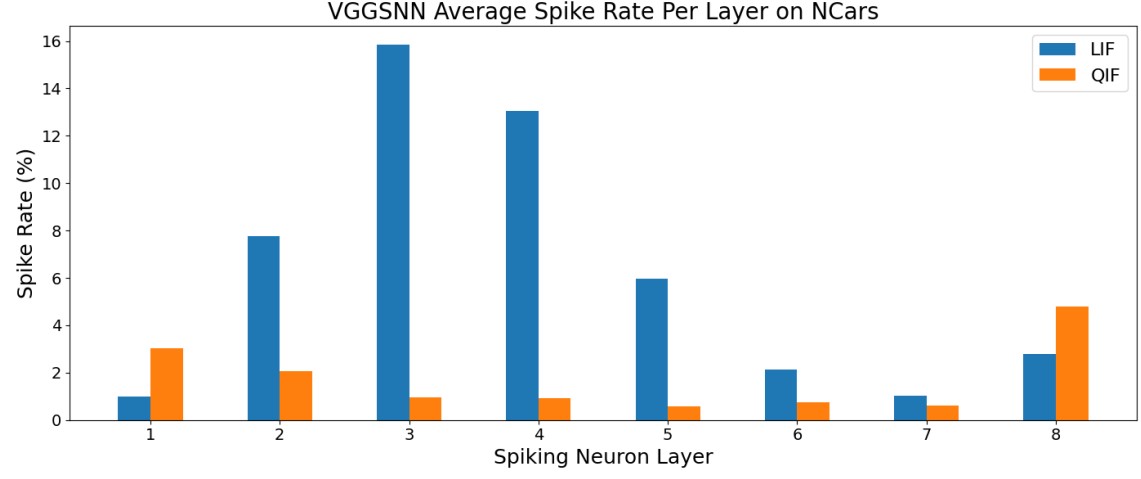

Figure 13: Average spike rate comparison of VGGSNN over the N-Cars validation set using QIF and LIF neuron models.

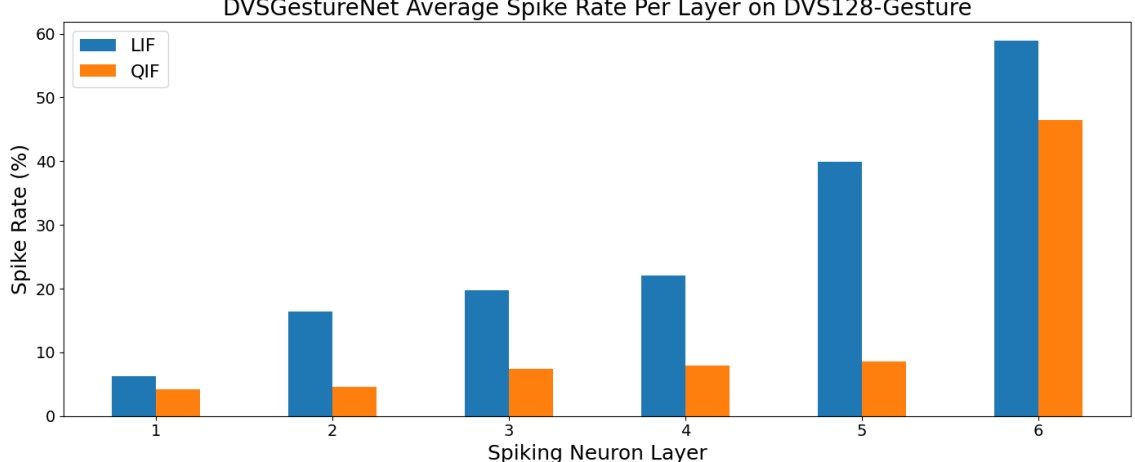

Figure 14: Average spike rate comparison of DVSGestureNet over the DVS128-Gesture validation set using QIF and LIF neuron models.

# H   LOSS CONTOUR PLOTS AND SURFACES

In this section, we present contour and surface plots of the loss landscape and training graphs for each model following the method described by Li et al. (2018). We train the same model architecture twice using LIF and QIF neurons.

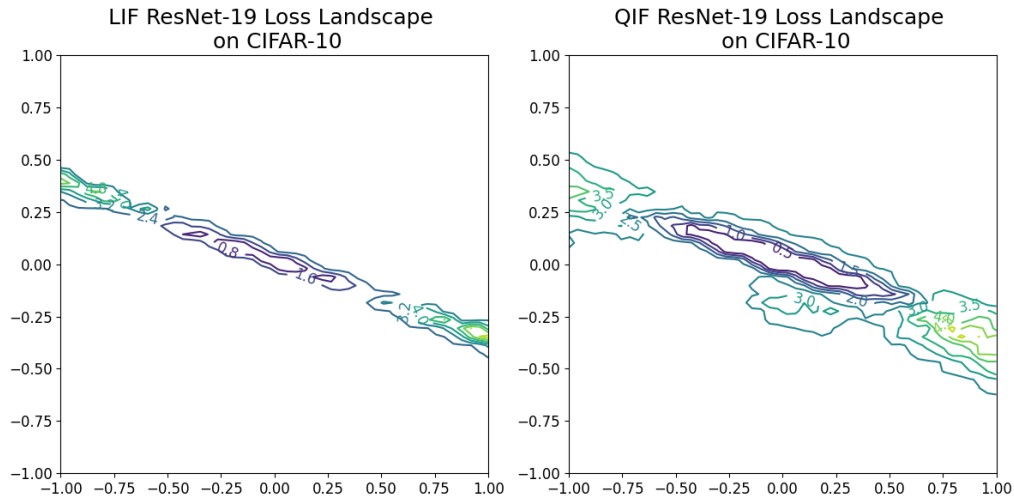

Figure 15: Post training loss landscape contour plot of ResNet-19 on CIFAR-10 using QIF and LIF neuron models.

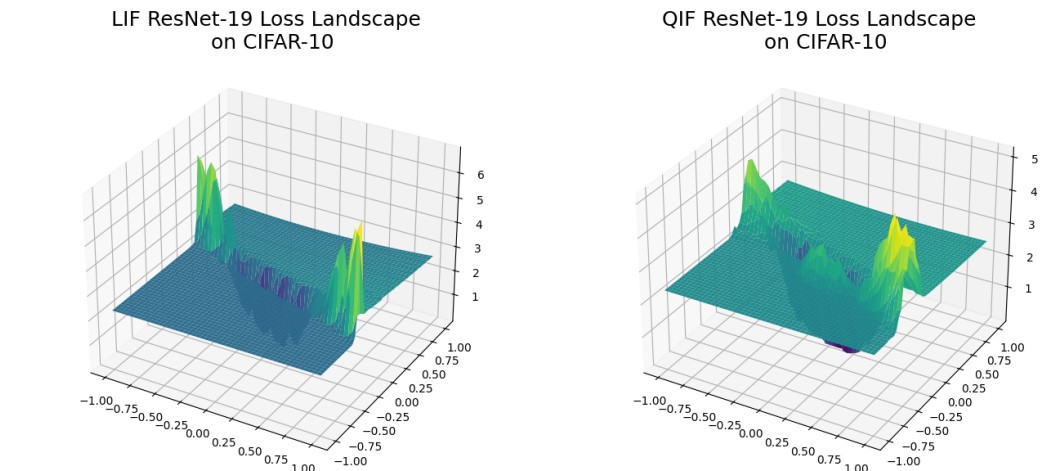

Figure 16: Post training loss surface of ResNet-19 over the CIFAR-10 validation set using QIF and LIF neuron models.

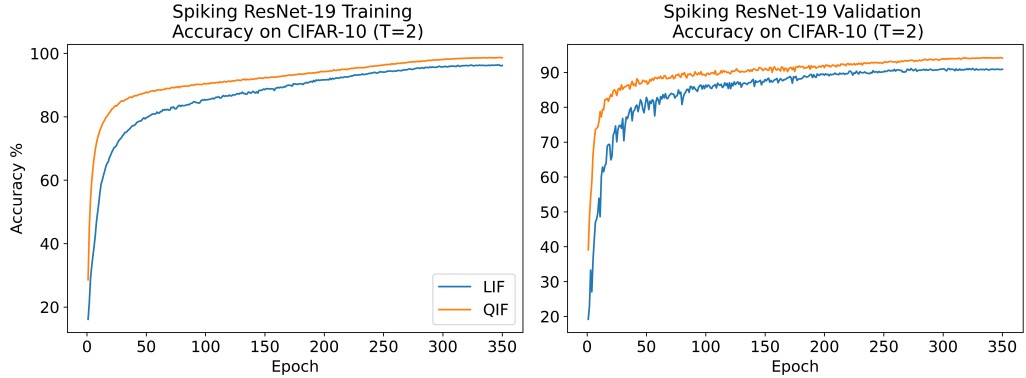

Figure 17: Training and validation accuracy comparison of ResNet-19 on CIFAR-10 using QIF and LIF neuron models.

Figures 15, 16, and 17 show a contour plot of loss surface, 3D visualization of the contour plot, and the training graphs of the QIF and LIF models trained on CIFAR-10 using ResNet-19.

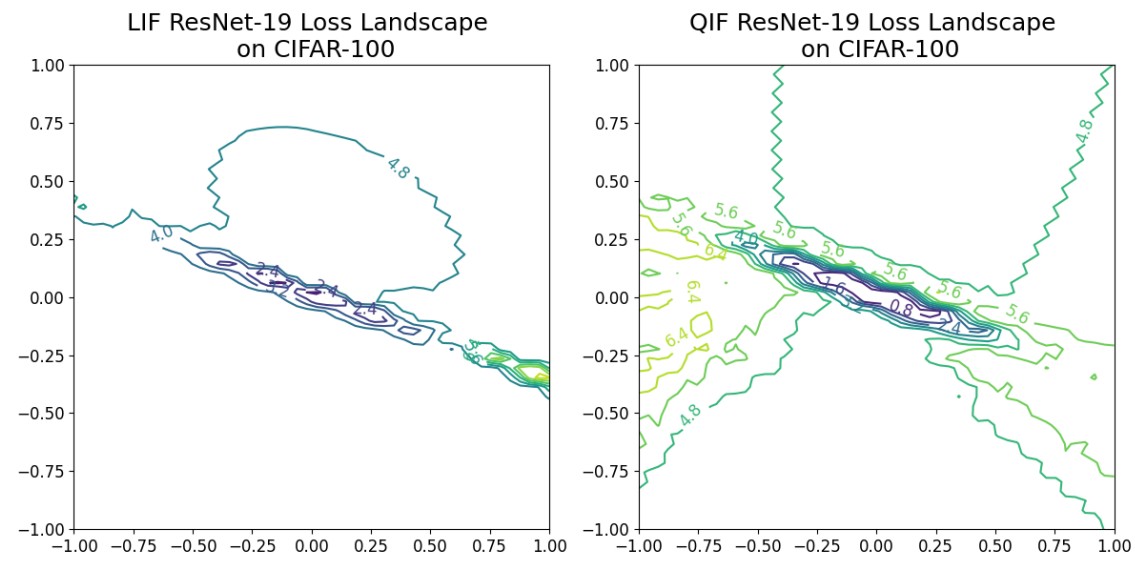

Figure 18: Post training loss landscape contour plot of ResNet-19 on CIFAR-100 using QIF and LIF neuron models.

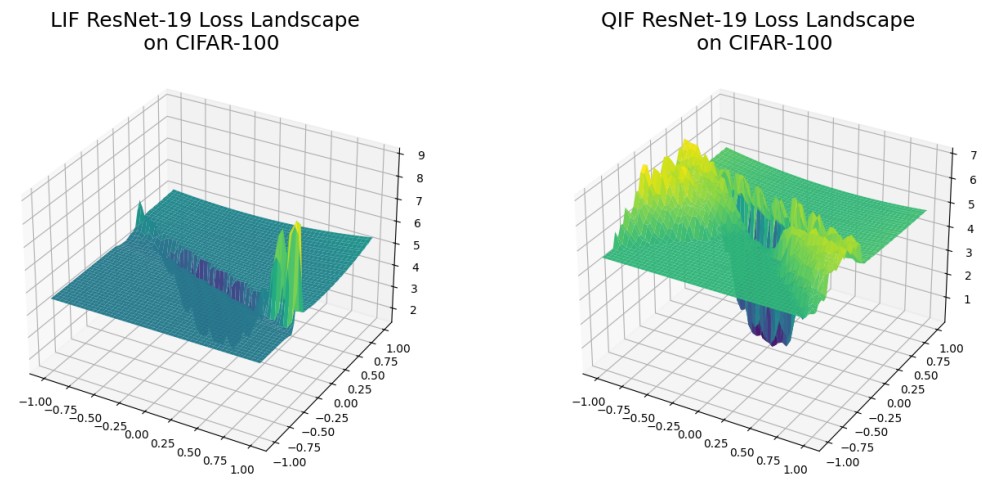

Figure 19: Post training loss surface of ResNet-19 on CIFAR-100 using QIF and LIF neuron models.

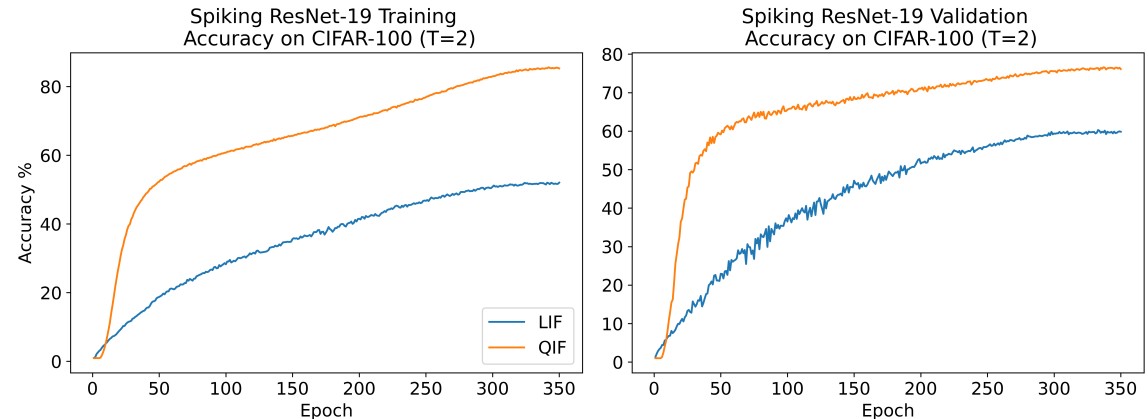

Figure 20: Training and validation accuracy comparison of ResNet-19 on CIFAR-100 using QIF and LIF neuron models.

Figures 18, 19, and 20 showcase the loss surfaces and training graphs of an LIF and QIF model trained on CIFAR-100. We see broader minima when using the QIF neuron model, which translates to superior training performance under the same conditions.

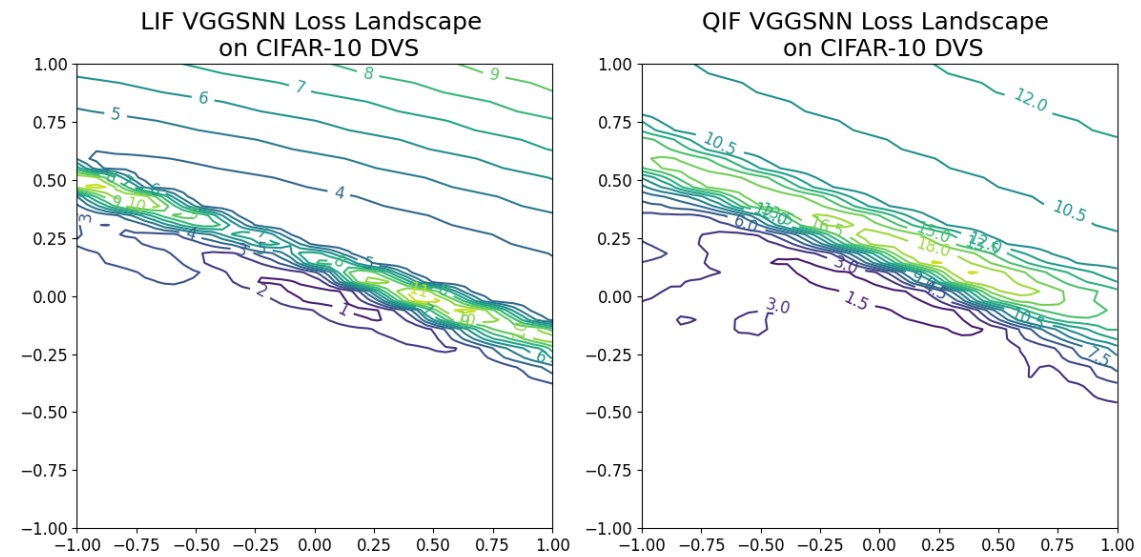

Figure 21: Post training loss landscape contour plot of VGGSNN on CIFAR-10 DVS using QIF and LIF neuron models.

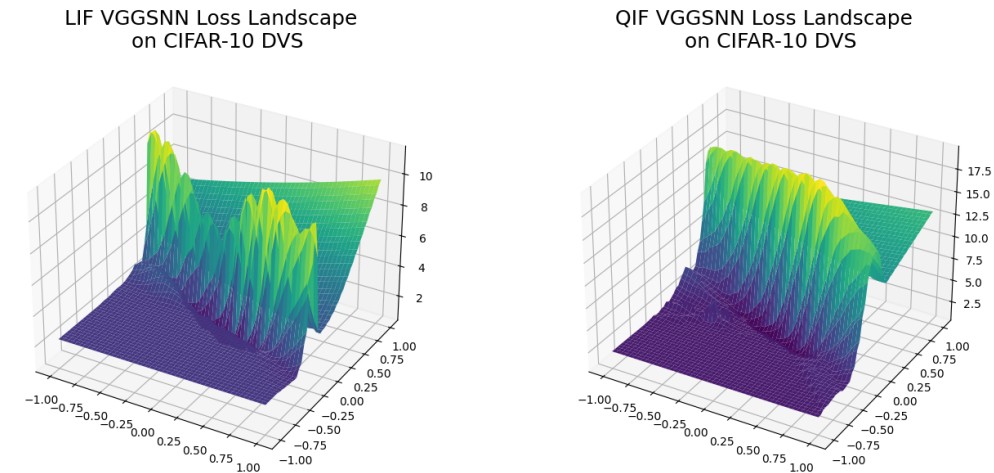

Figure 22: Post training loss surface of VGGSNN on CIFAR-10 DVS using QIF and LIF neuron models.

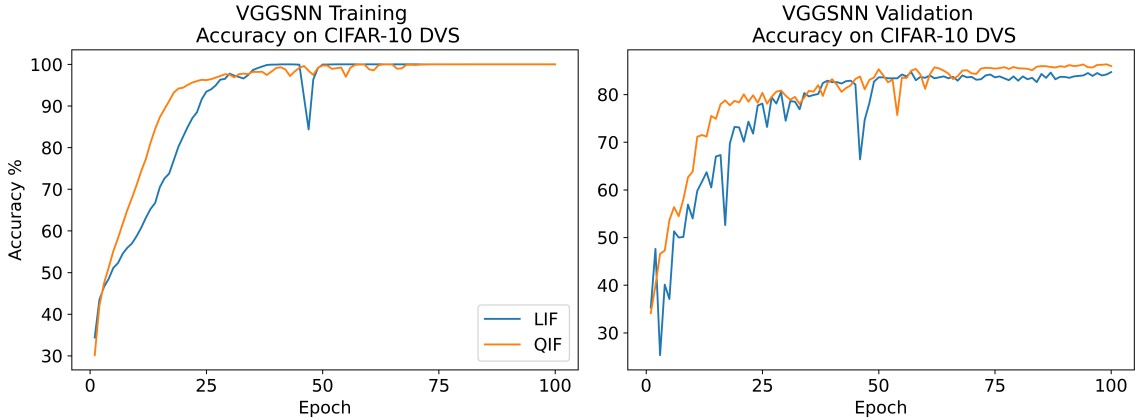

Figure 23: Training and validation accuracy comparison of VGGSNN on CIFAR-10 DVS using QIF and LIF neuron models.

Figures 21, 22, and 23 showcase the loss surfaces and training graphs of an LIF and QIF model trained on CIFAR-10 DVS. We see roughly the same size and shape minima for both neuron models, however, the LIF model's loss landscape is flatter. Both models obtain similar training trends, but the QIF model can generalize better.

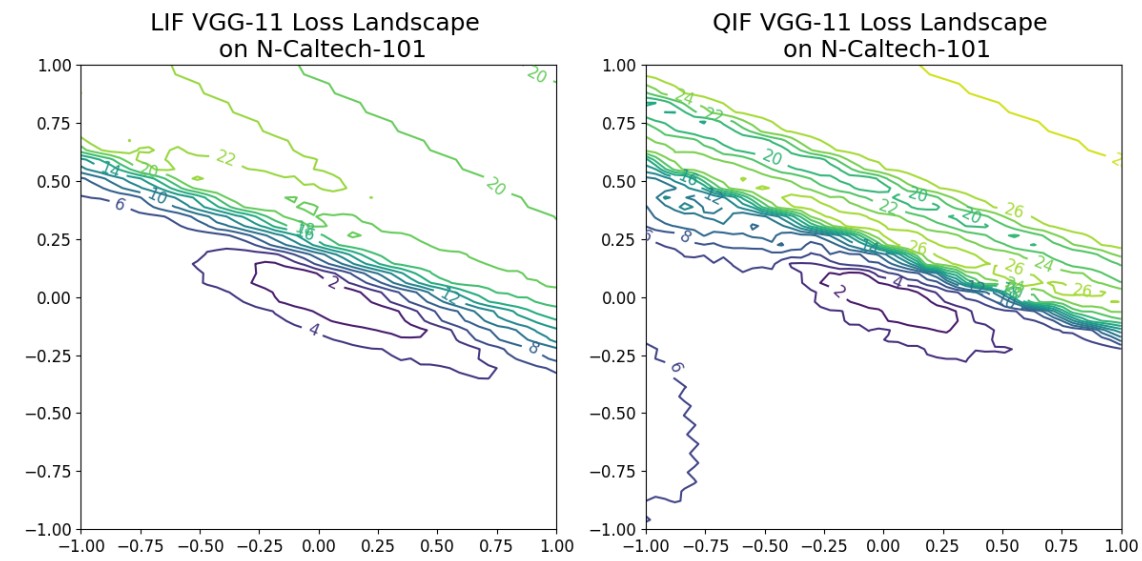

Figure 24: Post training loss landscape contour plot of VGG-11 on N-Caltech-101 using QIF and LIF neuron models.

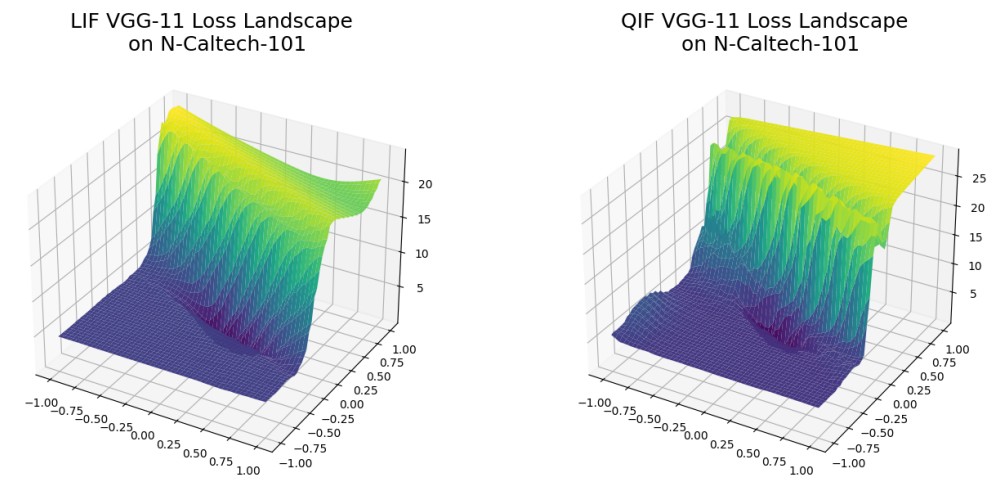

Figure 25: Post training loss surface of VGG-11 on N-Caltech-101 using QIF and LIF neuron models.

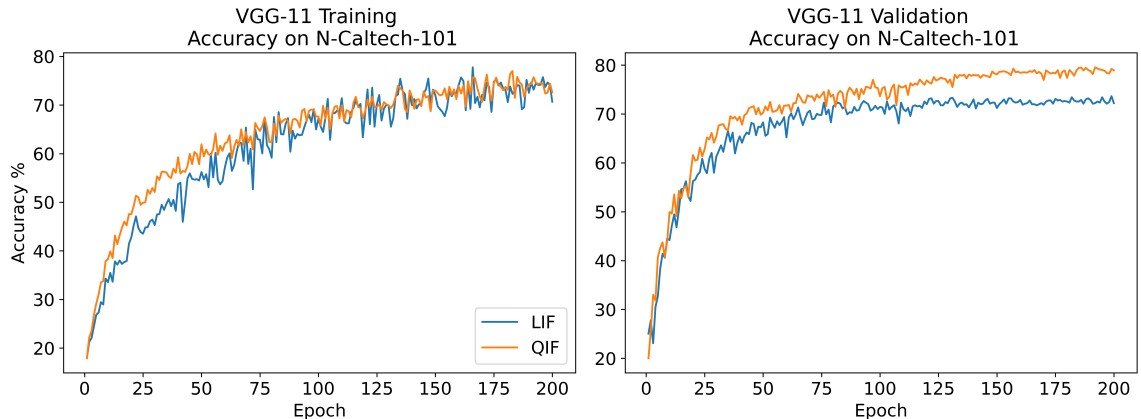

Figure 26: Training and validation accuracy comparison of VGG-11 on N-Caltech-101 using QIF and LIF neuron models.

Figures 24, 25, and 26 showcase the loss surfaces and training graphs of an LIF and QIF model trained on N-Caltech-101. Both loss surfaces have similar shapes similar to our results with CIFAR-10 DVS. When looking at the training graphs, we again see that the QIF model generalizes better and outperforms the LIF model by a significant margin.

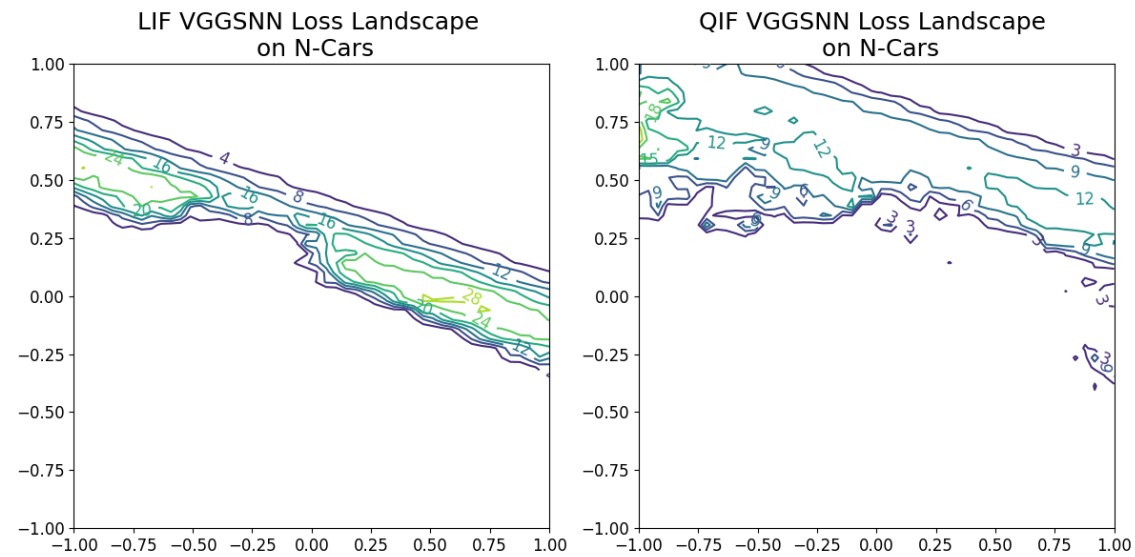

Figure 27: Post training loss landscape contour plot of VGGSNN on N-Cars using QIF and LIF neuron models.

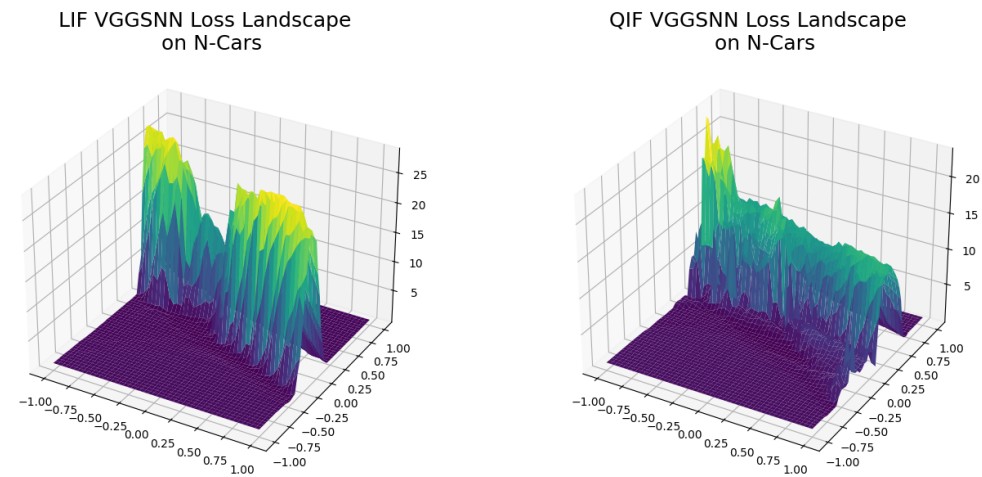

Figure 28: Post training loss surface of VGGSNN on N-Cars using QIF and LIF neuron models.

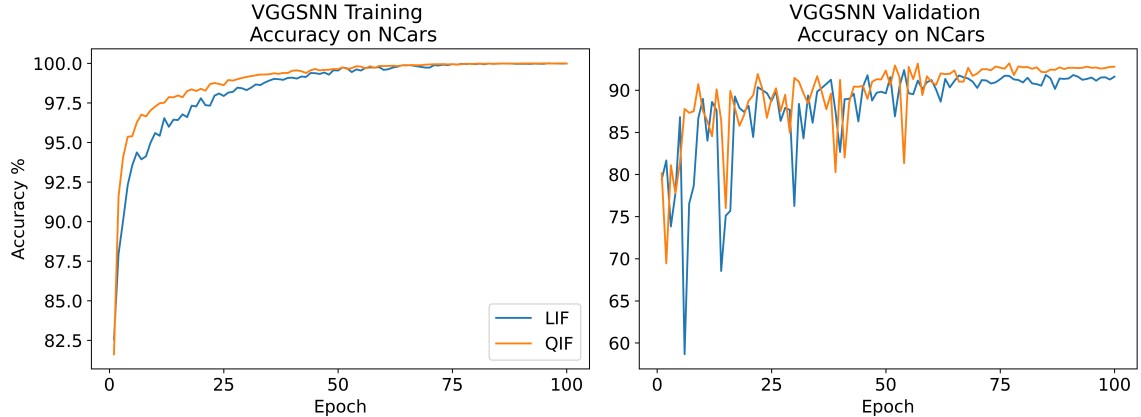

Figure 29: Training and validation accuracy comparison of VGGSNN on N-Cars using QIF and LIF neuron models.

Figures 27, 28, and 29 showcase the loss surfaces and training graphs of an LIF and QIF model trained on N-Cars. Both loss surfaces have many sharp peaks and valleys, with the QIF model producing many minima that are broader than those of the LIF model. The sharpness of both loss surfaces is reflected in the volatility of validation accuracy during training. The QIF model still manages to generalize better.

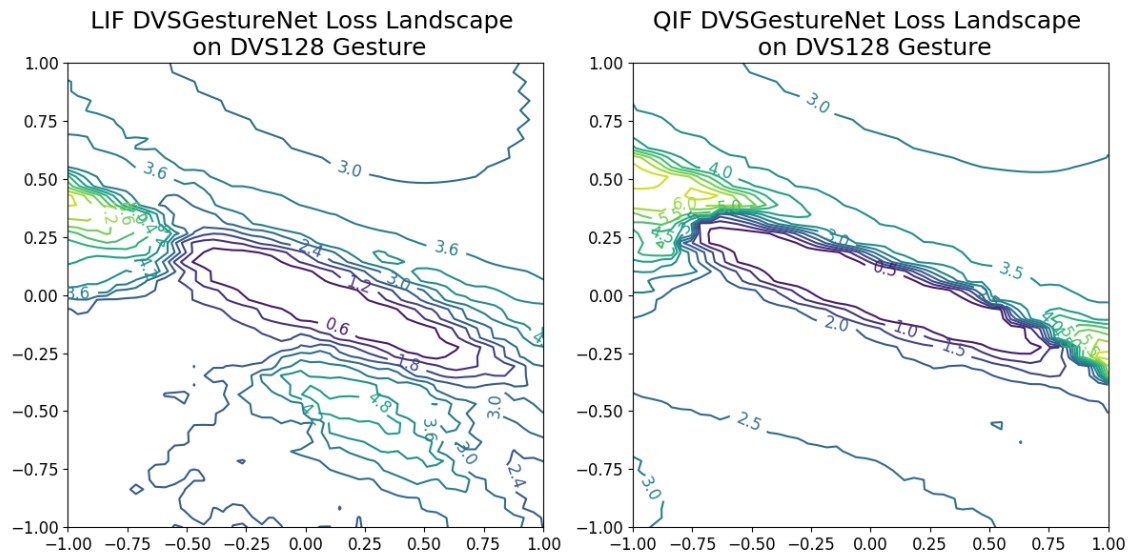

Figure 30: Post training loss landscape contour plot of DVSGestureNet on DVS128-Gesture using QIF and LIF neuron models.

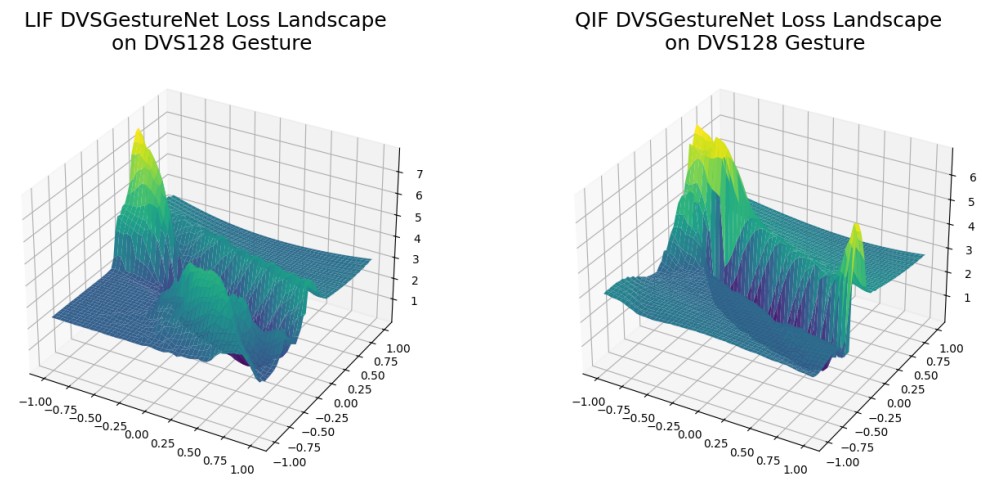

Figure 31: Post training loss surface of DVSGestureNet on DVS128-Gesture using QIF and LIF neuron models.

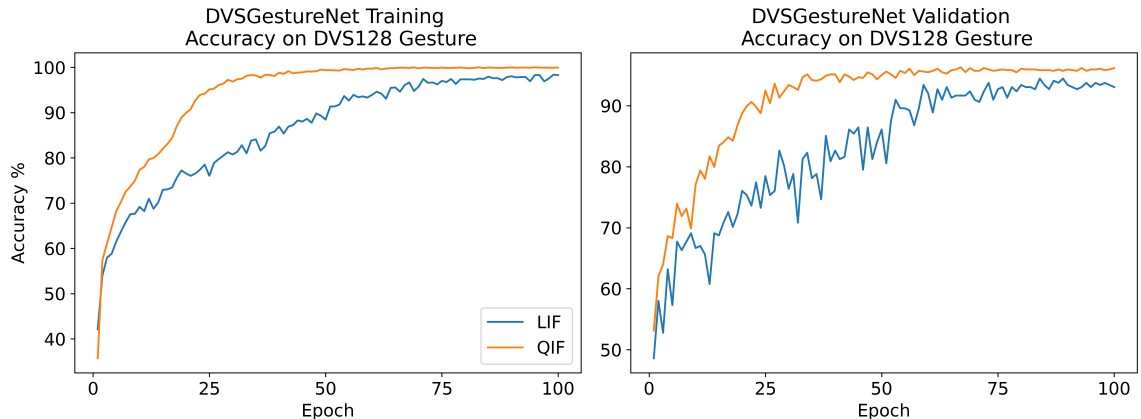

Figure 32: Training and validation accuracy comparison of DVSGestureNet on DVS128-Gesture using QIF and LIF neuron models.

Figures 30, 31, and 32 showcase the loss surfaces and training graphs of an LIF and QIF model trained on DVS128-Gesture. The QIF model's loss surface is overall much flatter than the LIF model with a smoother trajectory toward the minima. This is reflected in the training graphs, where the QIF model converges faster than the LIF model and generalizes better.

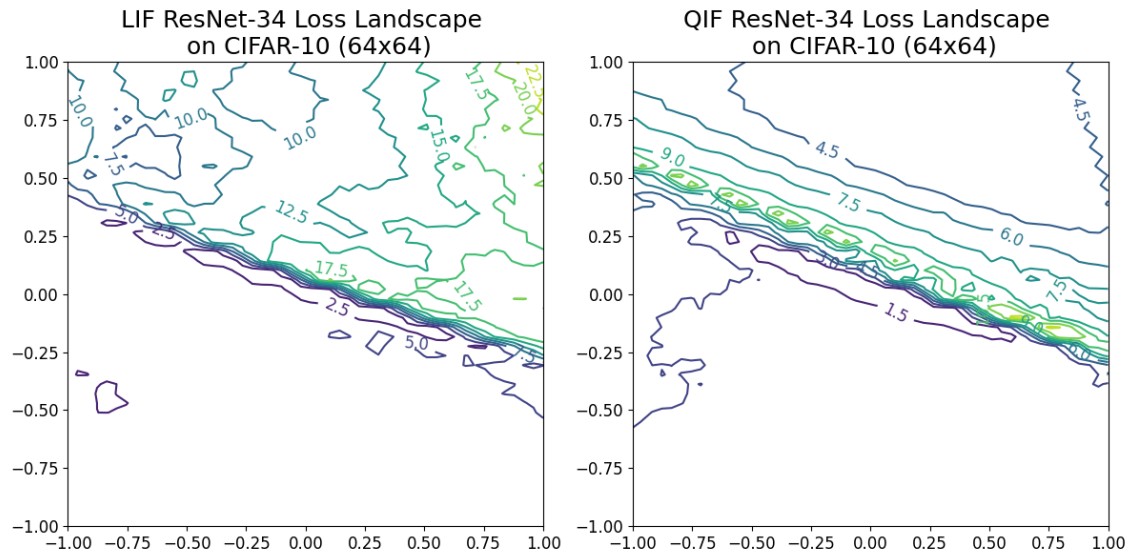

Figure 33: Post training loss landscape contour plot of ResNet-34 on CIFAR-10 (64x64) using QIF and LIF neuron models.

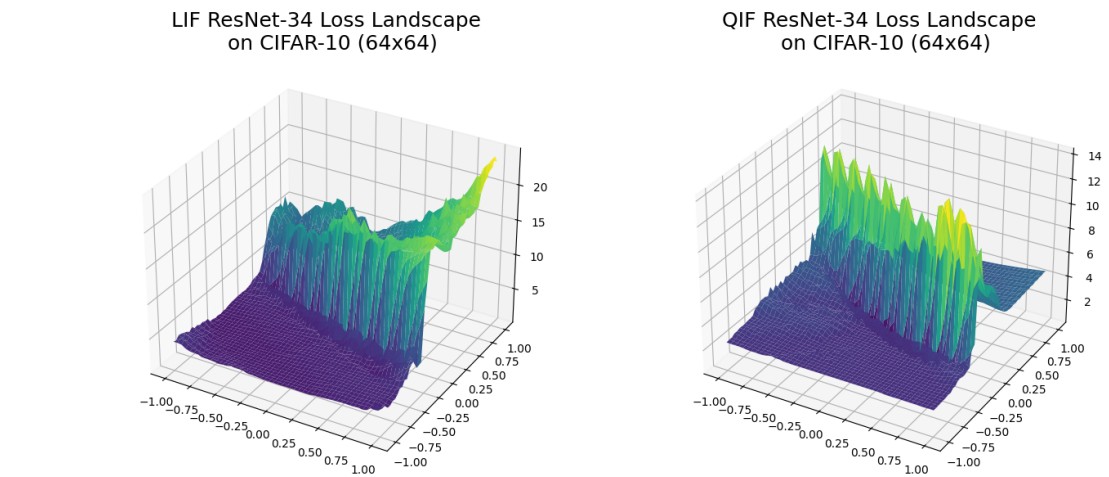

Figure 34: Post training loss surface of ResNet-34 on CIFAR-10 (64x64) using QIF and LIF neuron models.

Figures 33 and 34 showcase the loss surfaces of QIF and LIF models trained on CIFAR-10 (64x64). The QIF model has a sharper peak in the middle of its loss surface but has an overall flatter and smaller loss landscape than the LIF model. Additionally, the QIF loss landscape contains larger local minima.

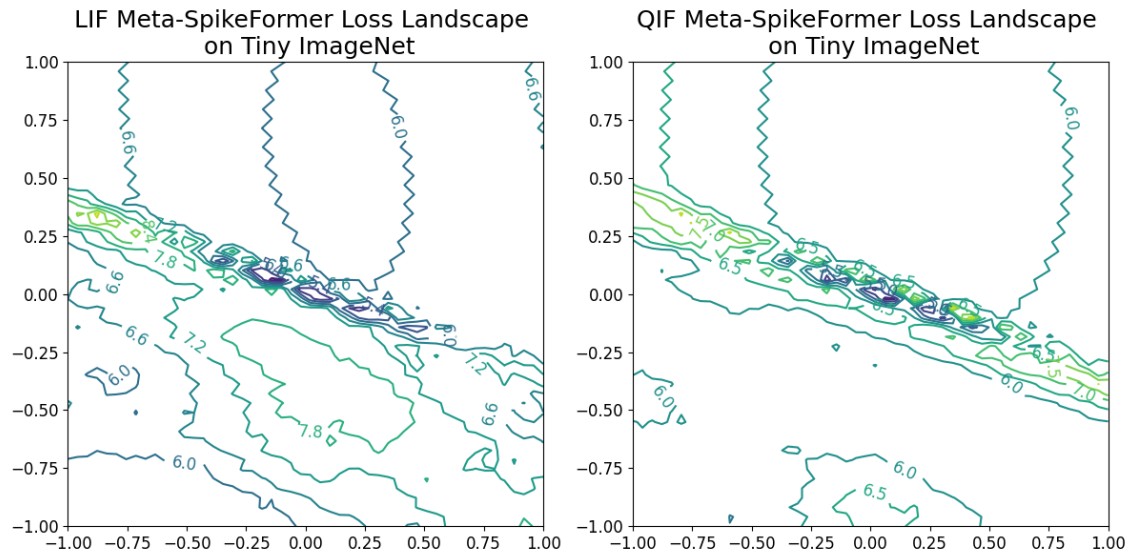

Figure 35: Post training loss landscape contour plot of Meta-SpikeFormer on Tiny ImageNet using QIF and LIF neuron models.

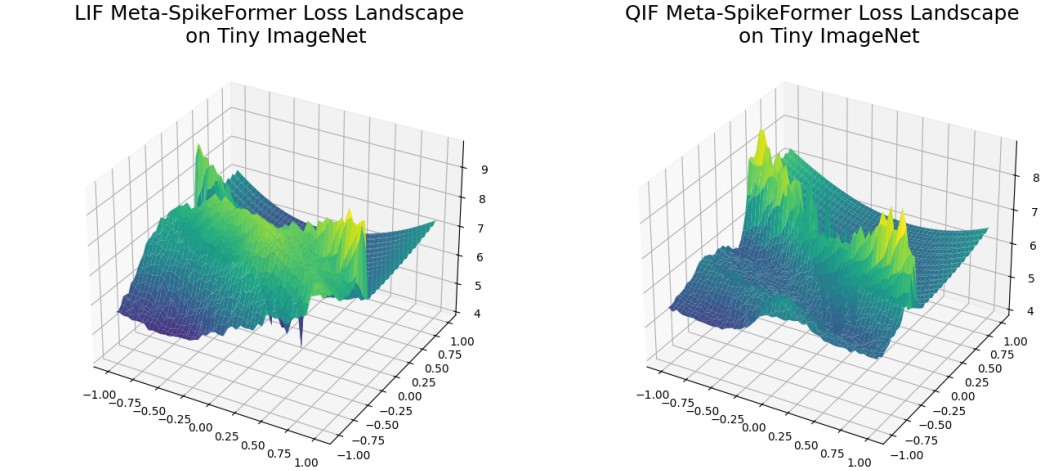

Figure 36: Post training loss surface of Meta-SpikeFormer on Tiny ImageNet using QIF and LIF neuron models.

Figures 35 and 36 showcase the loss surfaces of QIF and LIF models trained on Tiny ImageNet. The QIF model has larger local minima along with an overall flatter appearance. Both landscapes have sharp peaks.