# OpenReview forum: "Discretized Quadratic Integrate-and-Fire Neuron Model for Direct Training of Spiking Neural Networks"
_ICLR.cc/2025/Conference — ICLR 2025 Conference Withdrawn Submission_

### Official Review · Reviewer_hKm8 · 2024-11-01

**Soundness:** 3
**Presentation:** 2
**Contribution:** 2
**Rating:** 6
**Confidence:** 5

**Summary:**

This paper presents the discretized Quadratic Integrate-and-Fire (QIF) neuron model for training deep Spiking Neural Networks (SNNs). The QIF model is proposed as an alternative to the commonly used Leaky Integrate-and-Fire (LIF) neuron model, aiming to retain non-linear, biologically plausible dynamics after discretization. Through extensive evaluation, the paper demonstrates the QIF model's competitive performance on static datasets (CIFAR-10, CIFAR-100) and superior accuracy on several neuromorphic datasets (CIFAR-10 DVS, N-Caltech-101, N-Cars, DVS128-Gesture). Additionally, it shows that the QIF model improves energy efficiency by 20-123% compared to LIF neurons and offers faster convergence due to smoother loss landscapes and greater robustness to hyperparameter selection.

**Strengths:**

The introduction of a discretized QIF neuron model addresses a gap in the literature by capturing more complex, non-linear neuronal dynamics, potentially advancing SNNs closer to biological realism. The authors derive an equation to calculate surrogate gradient windows directly from QIF parameters, which minimizes issues with gradient mismatch and naive initialization. This analytical insight is a valuable addition to the SNN field. The model is extensively benchmarked on both static and neuromorphic datasets, demonstrating its generalizability and competitiveness with state-of-the-art methods. The focus on energy efficiency and robustness is particularly relevant for neuromorphic computing. The authors report significant energy efficiency improvements over LIF, along with smoother loss landscapes and faster convergence. This is crucial for the deployment of SNNs in real-world applications where power constraints are critical.

**Weaknesses:**

While the paper claims that the QIF model is more biologically plausible, it would benefit from a brief explanation of how the non-linear dynamics in QIF neurons more accurately reflect biological neurons, particularly compared to LIF. The reported 20-123% improvement in energy efficiency is broad, and it is unclear what factors influence this range. Clarifying whether architecture, dataset, or another factor contributes to this variation would improve interpretability. The claim of greater robustness to hyperparameter selection is promising, yet the paper would be strengthened by including more detailed ablation studies to substantiate this claim.

**Questions:**

The paper could address potential trade-offs between QIF and LIF neuron models, such as computational cost or latency, as these may be factors in deciding the appropriateness of QIF for specific applications.

---

> ### Author Response · Authors · 2024-11-21
>
> Hello Reviewer hKm8,
>
> We appreciate your time and effort in reviewing our work. Based on your insightful comments, we have modified the work and discussed our changes below.
>
> To address your concern, "The reported 20-123\% improvement in energy efficiency is broad, and it is unclear what factors influence this range. Clarifying whether architecture, dataset, or another factor contributes to this variation would improve interoperability., we added additional discussion within section 5.2 attempting to clarify theoretically which factors contribute most to the difference in energy efficiency. A snippet of this discussion is included below.
>
> ''Using Equation 17, we calculate the energy consumption of each neuron model in Table 1. We observe energy reduction ranging from 1.23 - 4.21x for the QIF neuron models. These savings are attributed to the non-linear dynamics of the QIF neuron, which tends to induce a voltage distribution with neurons further away from the threshold, as seen in Figure 9. These dynamics increase the difficulty for a neuron to spike, reducing the rate at which less important neurons may fire due to noise or low-quality features. Additionally, neuromorphic datasets show greater energy savings on average than static datasets. This difference may stem from the high sparsity and noise typical of these datasets that cause LIF models to follow noise and produce excess spikes while the QIF models handle this noise more effectively, reducing unnecessary spikes.'
>
> Note that we previously presented our energy efficiency gains using the percent difference formula, which measures the percentage two values differ from their mean. However, after a discussion with the team, we decided that the percent difference may provide misleading insights into our improvements. Therefore, we now present our results in terms of how many times better the QIF model is compared to the LIF model. We believe this new method helps to showcase our results in a more interpretable manner.
>
> We also added a discussion at the end of Section 5.2 regarding additional concerns that may come up about the computational cost/latency of implementing the QIF neuron model on neuromorphic and non-neuromorphic hardware. We hope this provides more insight into the trade-offs between the QIF and LIF neuron models and how the QIF model may overcome some of its limitations when implemented on neuromorphic hardware. A snippet of this discussion is included below.
>
> ''To discuss potential concerns related to computational complexity, we showcase the additional latency of a QIF neuron compared to a LIF neuron. A single LIF neuron requires one multiplication and one addition while a single QIF neuron requires three additions and two multiplications. Assuming the two additions required to compute $(u - u_{rest})$ and $(u - u_c)$, from Equation 11, can be done in parallel, the QIF neuron has one addition and multiplication more than the LIF neuron, leading to roughly 2x the computational complexity. On our non-neuromorphic experimental setup, we observe that this leads to inference time overheads between 1.04-1.39x, as shown in Table 2. Due to the limited public availability of neuromorphic hardware, it is difficult to calculate the exact computational overhead incurred by these additional operations. However, we do know that many neuromorphic hardware implementations, such as Intel Loihi 2 [1], follow event-driven paradigms. This means the lower spike rate of the QIF neuron has the potential to lower the computational overhead we observed on non-neuromorphic hardware. To put this in perspective, across all datasets and models, the QIF neuron produces an average of 45.47\% less spiking activity than the LIF neuron. Therefore, the QIF neuron will only require around half the number of active neurons during inference on average. This suggests that implementing these models on neuromorphic hardware can offset the additional computational complexity of the QIF neuron through its decreased spiking activity.'
>
> [1] Intel. Taking neuromorphic computing to the next level with Loihi 2 technology brief. https://www.intel.com/content/www/us/en/research/ neuromorphic-computing-loihi-2-technology-brief.html, 2021. Accessed: 03-19-2024.

---

> > ### Author Response · Authors · 2024-11-21
> >
> > To clarify exactly how the QIF neuron more accurately reflects biological neurons compared to the LIF neuron, we first added additional clarification in the difference of dynamics between the LIF and QIF neuron models throughout the entire paper. One example is at the start of Section 4.1, where we provide more background into neuron models as a whole, more details about the LIF neuron model, and discuss how both neuron models relate to biological neurons. A snippet of this discussion is included below.
> >
> > The Hodkin-Huxley (HH) neuron model was created to mimic the activity of neurons found within a giant squid and has proven itself invaluable in the field of neuroscience [3]. Over the years, simplifications of the HH neuron model have been introduced to reduce the computational complexity of its various equations and non-linear dynamics. The LIF neuron model is an extreme simplification that has proven itself to be a computationally efficient alternative. However, the LIF neuron model does not contain non-linear dynamics dependent on voltage as seen in the HH neuron model. We aim to bridge this gap by looking at other neuron models that contain non-linear dynamics without introducing large computational overhead. This initially led us to the Exponential Integrate-and-Fire (ExLIF) neuron model [3]. The ExLIF neuron model simplifies the HH neuron model and maintains much of its non-linear dynamics. However, due to the large computational cost of the ExLIF neuron, an approximation called the Quadratic Integrate-and-Fire (QIF) neuron model is often used in experimental settings [3].
> >
> > We hope this helps to better clarify the motivation and strengthen the narrative behind our work.
> >
> > Regarding your concern related to our claims related to hyperparameter robustness, we have included an additional experiment in Appendix F which further tests the hyperparameter robustness of the QIF neuron model. This additional experiment trains DVSGestureNet (containing over $10\times$ as many parameters as LeNet-5) on DVS128-Gesture and performs the same grid sweep of hyperparameters as we did for the LeNet-5 experiment. For each set of hyperparameters, we train the DVS128-Gesture model for 20 epochs. We chose 20 epochs as a trade-off due to the computation time needed to train the 150 separate models required for the hyperparameter sweep and the short rebuttal period. Our results showcase the QIF neuron model outperforms the LIF neuron model in terms of minimum, maximum, and average validation accuracy. Additionally, the QIF neuron model exhibits a much smaller variance in model performance. We believe that this additional experiment showcases the QIF neuron model's hyperparameter robustness to scale for more complex tasks.
> >
> > Thank you again for your comments. If you have any additional questions, comments, or concerns please let us know and we will address them as soon as possible.
> >
> > [3] Wulfram Gerstner, Werner M. Kistler, Richard Naud, and Liam Paninski. Neuronal Dynamics: From Single Neurons to Networks and Models of Cognition. Cambridge University Press, 2014.

---

### Official Review · Reviewer_U6Ka · 2024-11-01

**Soundness:** 2
**Presentation:** 3
**Contribution:** 2
**Rating:** 6
**Confidence:** 3

**Summary:**

This paper proposed a new spiking neuronal model to overcome the shortcomings of previous discretized neuron model. The method is widely validated on several datasets and compared with other methods.

**Strengths:**

1. The methods are validated on two image recognition datasets and several neuromorphic datasets, which make good contributions. Though ImageNet results are not presented, it can be understandable that surrogate gradient learning on such dataset is very expensive.

2. The loss landscape is visualized in the supplementary materials, which is very helpful for better understanding the method

**Weaknesses:**

The context and motivation are very unclear.

The paper mentions that a technique is used before it is first proposed. See questions for details. The author should explain this.

**Questions:**

1. Line 41 to Line 45. Could you explain what "discretization techniques" really mean? Also, it is confusing that TrueNorth used such a technique in 2015 but the paper said this technique was introduced in 2018. Could you explain in detail how this happens?

2. Line 207 "Therefore, we introduce our discretized QIF neuron model". The current story is that the "discretization" of the current spiking neuron model is not good so you introduced a new one. However, the proposed neuronal model is still discretized. This makes the motivation of this paper not clear. Could you explain more about the motivation of your method?


--------------------------
Updates after rebuttal
The author responded to the two questions I raised and addressed them. Thus, I increase my score from 5 to 6.

---

> ### Author Response · Authors · 2024-11-21
>
> Hello Reviewer U6Ka,
>
> We appreciate your time and effort in reviewing our work. Based on your insightful comments, we have modified the work and discussed our changes below.
>
> Regarding your question about "what discretization techniques mean". Discretization techniques, in our context of ordinary differential equations, relate to the technique used to convert a differential equation from continuous time to discrete time steps. There are various techniques to do this, such as finite difference, finite element, finite volume, etc. However, in our work, when we reference other discretization techniques, we are referencing alternative finite difference techniques to use instead of Euler's method, such as Runge-Kutta, Adams-Bashforth, Adams-Moultan, etc. In response to this concern, we have removed this language from the introduction to not cause any confusion for future readers. We hope this answers your question and provides clarification.
>
> Regarding the second part of your question related to Lines 41-45, thank you for catching our mistake. The writing in that section was unclear and we hope to have addressed this issue in the new revision. The TrueNorth platform was released in 2015, while the work implementing always-on speech recognition was published in 2017. Before, we had previously insinuated that the always-on speech recognition work utilized the discretized LIF neuron model which was introduced in 2018 by Wu et al. [2]. This was a large mistake in our writing and does not make sense chronologically. We were trying to use these past examples of real applications to motivate Spiking Neural Networks on neuromorphic hardware. In this case, we didn't want the reader to focus on what neuron model work was using yet. However, as you pointed out, our previous writing suggested that both these works used the discretized LIF neuron model. To clarify this issue, we updated this section in the introduction to separate these examples on neuromorphic hardware and the discretized LIF neuron model to avoid future confusion. Thank you again for pointing out this mistake.
>
> For your second question regarding Line 207 and onwards, we have significantly reworked this section to provide more motivation and background for our work. These changes were made at the top of Section 4.1. To address your question deeper here, the discretization process itself is not at fault, rather the nature of the LIF neuron is. The LIF neuron's voltage decays linearly in direct proportion to its voltage. This behavior is seen in both the ordinary differential equation and the discretized form of the LIF neuron. On the other hand, the QIF neuron's voltage decays non-linearly in relation to the magnitude of its voltage. Again, these dynamics are seen in both the ordinary differential equation and the discretized form of the QIF neuron. Your comment has helped bring this mistake to our attention and we have clarified this fact throughout the paper. We believe these changes have helped to give our work more context and strengthen the motivation and narrative behind it.
>
> Additionally, based on the great comments and feedback of other reviewers, we would like to let you know that we have made notable changes to other sections of our work. To summarize these changes, in Section 5.2, we have added additional discussion on the hardware overhead of the QIF neuron model and more reasoning on how the QIF neuron model can decrease energy consumption. In Section 5.4, we add more details on how the QIF neuron model seems to facilitate faster model convergence. We added Appendix E which includes additional preliminary experiments with larger model architectures such as ResNet-34 trained on CIFAR-10 as well as a spiking transformer architecture trained on Tiny ImageNet.
>
> Thank you again for your comments. If you have any additional questions, comments, or concerns please let us know and we will address them as soon as possible.
>
> [2] Yujie Wu, Lei Deng, Guoqi Li, Jun Zhu, and Luping Shi. Spatio-temporal backpropagation for training high-performance spiking neural networks. Frontiers in Neuroscience, 12, 2018. ISSN 1662-453X. doi: 10.3389/fnins.2018.00331. URL https://www.frontiersin.org/journals/neuroscience/articles/10.3389/fnins.2018.00331

---

> > ### Comment · Reviewer_U6Ka · 2024-11-21
> >
> > Thanks for your replies to my questions. Appreciate it.
> >
> > I have raised my score from 5 to 6

---

### Official Review · Reviewer_V8Tp · 2024-11-03

**Soundness:** 2
**Presentation:** 2
**Contribution:** 2
**Rating:** 3
**Confidence:** 5

**Summary:**

The paper introduces a discretized Quadratic Integrate-and-Fire (QIF) neuron model for SNNs to better preserve the complex non-linear dynamics of biological neurons compared to the LIF model. This enhanced model addresses issues like reduced performance and increased energy consumption caused by extraneous spiking activity in LIF-based SNNs.

**Strengths:**

1. Significance: The paper addresses the critical issue of energy consumption in neural networks by introducing a novel QIF neuron model. This focus on reducing energy usage is highly relevant and contributes meaningfully to the SNN community.

2. Clarity: The paper is well-written and easy to understand, with clear explanations of the QIF model’s formulation.

**Weaknesses:**

1. Limited Experimental Validation: The study primarily employs a single, outdated neural network architecture, lacking evaluations on more modern and diverse models such as Transformers. Additionally, experiments do not cover a range of model sizes, which restricts the assessment of the QIF model's generalizability and scalability.

2. Absence of Large-Scale Dataset: The paper does not include evaluations on the ImageNet dataset.

3. Increased Hyperparameter Complexity: Compared to the traditional LIF model, the QIF model introduces additional hyperparameters.  Although parameter insensitivity is claimed, the supporting evidence relies heavily on simple architectures and datasets, which is not convincing.

4. Marginal Performance Improvements: While the QIF model achieves significant energy reductions, its performance gains in terms of accuracy are not consistently superior to existing methods. This raises questions about whether the additional complexity and parameter overhead are justified by the modest performance enhancements.

5. Lack of Neuromorphic Hardware Adaptability Discussion: The paper does not discuss the compatibility and adaptability of the QIF neuron model with neuromorphic hardware.

**Questions:**

1. Could you provide a more detailed sensitivity analysis of the additional parameters introduced in the QIF model?

2. Could you provide insights into the theoretical foundations that explain the improved energy efficiency?

3. What are the potential solutions for hardware deployment?

4. Please provide all the experiments I mentioned in the Weakness to demonstrate the effectiveness.

---

> ### Author Response · Authors · 2024-11-21
>
> Hello Reviewer V8Tp,
>
> We appreciate your time and effort in reviewing our work. Based on your insightful comments, we have modified the work and discussed our changes below.
>
> To tackle the weakness related to Limited Experimental Validation and Absence of Large Scale Dataset, we have included two additional experiments within Appendix E of our work. The first experiment involves training a ResNet-34 model using CIFAR-10, scaling the images to $64\times64$. ResNet-34 has around $2 \times$ the parameters as ResNet-19. In this experiment, we observe a substantial energy improvement from the QIF neuron model of $2.15 \times$ less than the LIF neuron model and competitive accuracy between both models. We hope this experiment better showcases the generalizability of the QIF neuron to larger CNN architectures. In our second experiment, we use a Spiking Vision Transformer model and the Tiny ImageNet dataset to showcase the QIF neuron model on a non-CNN architecture and more complex task. We choose the Tiny ImageNet Dataset over ImageNet due to the time constraints of the rebuttal process. However, we feel that Tiny ImageNet provides useful insight due to its increased complexity and similarity to ImageNet. We are still in the process of collecting results with this model and dataset but would like to share our preliminary findings. The QIF model consumes approximately $8.20 mJ$ while the LIF neuron model consumes $.99\times$ less at $8.14 mJ$. In terms of accuracy, we initially see the LIF model outperform the QIF model. The QIF model catches back up and surpasses the LIF model later in training obtaining validation accuracies of $39.16$\% and $38.16$\% respectively. These accuracies are rather low as we only used two timesteps and did not use augmentations such as cutmix/mixup or label smoothing. We used two timesteps due to the computational cost of larger timesteps and the short rebuttal period. We didn't use additional augmentations so we can better attribute the differences in performance to the neuron model rather than the augmentations. Figures showcasing the training process and spike rates of each model are included in Appendix E. We hope his experiment showcases the QIF neuron model's applicability to other model architectures as well as more complex datasets. It remains an interesting research direction for us to explore the applicability of the QIF neuron model to tasks other than image classification as well as other network architectures.
>
> Regarding your concern related to Increased Hyperparameter Complexity, we have included an additional experiment in Appendix F which further tests the hyperparameter robustness of the QIF neuron model. This additional experiment trains DVSGestureNet (containing over $10\times$ as many parameters as LeNet-5) on DVS128-Gesture and performs the same grid sweep of hyperparameters as we did for the LeNet-5 experiment. For each set of hyperparameters, we train the DVS128-Gesture model for 20 epochs. We chose 20 epochs as a trade-off due to the computation time needed to train the 150 separate models required for the hyperparameter sweep and the short rebuttal period. Our results showcase the QIF neuron model outperforms the LIF neuron model in terms of minimum, maximum, and average validation accuracy. Additionally, the QIF neuron model exhibits a much smaller variance in model performance. We believe that this additional experiment showcases the QIF neuron model's hyperparameter robustness to scale for more complex tasks.

---

> > ### Author Response · Authors · 2024-11-21
> >
> > For the Marginal Performance Improvements and Lack of Neuromorphic Hardware Adaptability, we agree that the QIF neuron model has marginal accuracy gains that are not consistent across all models and datasets. To address your concerns that the additional complexity and parameter overhead may not be justified due to these marginal accuracy gains, we have included various changes in Section 5.2. First, we swapped the order in which we presented accuracy and energy-efficiency results. We aimed to present the QIF neuron model as a more energy-efficient alternative to the LIF neuron model. We feel this was lost in the narrative and we have updated the narrative to match this better. Next, we change the way we present our energy efficiency results. We previously used the percent difference formula, which measures the percentage two values differ from their mean. However, after a discussion with the team, we decided that the percent difference may provide misleading insights into our improvements. Therefore, we now present our results in terms of how many times better the QIF model is compared to the LIF model. We believe this showcases our results in a more interpretable manner. Lastly, we added additional discussion on the computational overhead of the QIF neuron model on both neuromorphic and non-neuromorphic hardware. We first showcase the additional overhead of the QIF neuron model on non-neuromorphic hardware. Then we discuss how neuromorphic hardware can potentially mitigate this overhead due to the significant decrease in spiking activity seen when using the QIF neuron model. A snippet of this section is included.
> >
> > ''To discuss potential concerns related to computational complexity, we showcase the additional latency of a QIF neuron compared to a LIF neuron. A single LIF neuron requires one multiplication and one addition while a single QIF neuron requires three additions and two multiplications. Assuming the two additions required to compute $(u - u_{rest})$ and $(u - u_c)$, from Equation 11, can be done in parallel, the QIF neuron has one addition and multiplication more than the LIF neuron, leading to roughly 2x the computational complexity. On our non-neuromorphic experimental setup, we observe that this leads to inference time overheads between 1.04-1.39x, as shown in Table 2. Due to the limited public availability of neuromorphic hardware, it is difficult to calculate the exact computational overhead incurred by these additional operations. However, we do know that many neuromorphic hardware implementations, such as Intel Loihi 2 [1], follow event-driven paradigms. This means the lower spike rate of the QIF neuron has the potential to lower the computational overhead we observed on non-neuromorphic hardware. To put this in perspective, across all datasets and models, the QIF neuron produces an average of 45.47\% less spiking activity than the LIF neuron. Therefore, the QIF neuron will only require around half the number of active neurons during inference on average. This suggests that implementing these models on neuromorphic hardware can offset the additional computational complexity of the QIF neuron through its decreased spiking activity.'
> >
> > Thank you again for your comments. If you have any additional questions, comments, or concerns please let us know and we will address them as soon as possible.
> >
> > [1] Intel. Taking neuromorphic computing to the next level with Loihi 2 technology brief. https://www.intel.com/content/www/us/en/research/ neuromorphic-computing-loihi-2-technology-brief.html, 2021. Accessed: 03-19-2024.

---

> > > ### Comment · Reviewer_V8Tp · 2024-12-02
> > >
> > > I sincerely thank the authors for their thorough response. However, after a careful review of the feedback and other reviewers' opinions, I think that the QIF model still requires additional experimental validation on large-scale datasets and hyperparameter studies. As such, I maintain my original score.

---

> > > > ### Author Response · Authors · 2024-12-02
> > > >
> > > > Dear Reviewer V8Tp,
> > > >
> > > > We would like to thank you again for the thoughts and insights you provided us with and the help they have given us in improving our work.

---

### Official Review · Reviewer_tkRP · 2024-11-03

**Soundness:** 2
**Presentation:** 3
**Contribution:** 2
**Rating:** 5
**Confidence:** 5

**Summary:**

This paper proposes replacing the traditional Leaky Integrate-and-Fire (LIF) neuron model with a Quadratic Integrate-and-Fire (QIF) neuron model in spiking neural networks (SNNs). The authors argue that QIF neurons retain more complex nonlinear dynamics that may better approximate biological neuron behavior. They introduce a custom surrogate gradient window to stabilize the training of QIF neurons and conduct experiments on multiple datasets to demonstrate potential advantages. However, the paper lacks extensive theoretical analysis and validation on larger datasets such as ImageNet and more complex network architectures (more deeper Spiking ResNet and Spiking Transformer). The experimental scope is limited, and there is minimal discussion on hardware implementation challenges, making the current contribution modest.

**Strengths:**

The paper makes a meaningful attempt to address the limitations of the traditional Leaky Integrate-and-Fire (LIF) neuron model by introducing the Quadratic Integrate-and-Fire (QIF) neuron model, which retains more complex nonlinear dynamics that may better approximate biological neurons. This choice is theoretically promising, as it could enhance the modeling capabilities of spiking neural networks (SNNs) by introducing more biologically realistic dynamics. Additionally, the paper introduces a surrogate gradient window tailored to QIF neurons, a practical contribution that improves training stability in spiking neural networks, and presents experimental results on multiple datasets showcasing QIF's potential benefits.

**Weaknesses:**

The paper lacks theoretical depth and sufficient experimental validation for QIF neurons in more complex scenarios. The primary contribution seems limited to replacing LIF neurons with QIF neurons and conducting experiments on small-scale datasets with simpler models. There is no comprehensive analysis of the impact of QIF-specific parameters (such as the resting potential and critical spiking threshold) on network performance or stability. Furthermore, the experiments are limited without testing on larger datasets like ImageNet or deeper, more complex architectures such as Transformers or deeper Spiking ResNet. These limitations make it challenging to assess the broader applicability and stability of QIF neurons in practical scenarios, and the hardware deployment considerations for QIF neurons are not adequately addressed.

**Questions:**

1.	Could the authors provide a more comprehensive analysis of QIF-specific parameters, such as the resting potential, critical spiking threshold, and decay rate? How do these parameters affect stability and convergence in different scenarios?
2.	Are there any plans to extend the experiments to larger datasets, such as ImageNet, or to more complex network architectures like Spiking ResNet or Transformers? This would significantly strengthen the paper’s claims about the general applicability of QIF neurons.
3.	How does the QIF neuron model compare to LIF in terms of computational and energy efficiency, especially in the context of neuromorphic hardware? Since QIF introduces more complex dynamics, it might face practical limitations when deployed on resource-constrained hardware.

---

> ### Author Response · Authors · 2024-11-21
>
> Hello Reviewer tkRP,
>
> We appreciate your time and effort in reviewing our work. Based on your insightful comments, we have modified the work and discussed our changes below.
>
> To address your first question, we have included an additional experiment in Appendix F which further tests the hyperparameter robustness of the QIF neuron model. This additional experiment trains DVSGestureNet (containing over $10\times$ as many parameters as LeNet-5) on DVS128-Gesture and performs the same grid sweep of hyperparameters as we did for the LeNet-5 experiment. For each set of hyperparameters, we train the DVS128-Gesture model for 20 epochs. We chose 20 epochs as a trade-off due to the computation time needed to train the 150 separate models required for the hyperparameter sweep and the short rebuttal period. Our results showcase the QIF neuron model outperforms the LIF neuron model in terms of minimum, maximum, and average validation accuracy. Additionally, the QIF neuron model exhibits a much smaller variance in model performance. We believe that this additional experiment showcases the QIF neuron model's hyperparameter robustness to scale for more complex tasks.
>
> To address your question related to plans to extend our experiments to larger datasets and other model architectures, we have included two additional experiments in Appendix E. The first experiment involves training a ResNet-34 model using CIFAR-10, scaling the images to $64\times64$. ResNet-34 has around $2 \times$ the parameters as ResNet-19. In this experiment, we observe a substantial energy improvement from the QIF neuron model of $2.15 \times$ less than the LIF neuron model and competitive accuracy between both models. We hope this experiment better showcases the generalizability of the QIF neuron to larger CNN architectures. In our second experiment, we use a Spiking Vision Transformer model and the Tiny ImageNet dataset to showcase the QIF neuron model on a non-CNN architecture and more complex task. We choose the Tiny ImageNet Dataset over ImageNet due to the time constraints of the rebuttal process. However, we feel that Tiny ImageNet provides useful insight due to its increased complexity and similarity to ImageNet. We are still in the process of collecting results with this model and dataset but would like to share our preliminary findings. The QIF model consumes approximately $8.20 mJ$ while the LIF neuron model consumes $.99\times$ less at $8.14 mJ$. In terms of accuracy, we initially see the LIF model outperform the QIF model. The QIF model catches back up and surpasses the LIF model later in training obtaining validation accuracies of $39.16$\% and $38.16$\% respectively. These accuracies are rather low as we only used two timesteps and did not use augmentations such as cutmix/mixup or label smoothing. We used two timesteps due to the computational cost of larger timesteps and the short rebuttal period. We didn't use additional augmentations so we can better attribute the differences in performance to the neuron model rather than the augmentations. Figures showcasing the training process and spike rates of each model are included in Appendix E. We hope his experiment showcases the QIF neuron model's applicability to other model architectures as well as more complex datasets. It remains an interesting research direction for us to explore the applicability of the QIF neuron model to tasks other than image classification as well as other network architectures.

---

> > ### Author Response · Authors · 2024-11-21
> >
> > To discuss potential computational overheads and energy efficiency, we have reworked Section 5.2. First, we provide a more in-depth discussion of the energy reductions seen when using the QIF neuron model and provide additional reasoning using both the QIF equation and experimental results for how the QIF model achieves this reduction. We have included a snippet of the revised section here.
> >
> > ''Using Equation 17, we calculate the energy consumption of each neuron model in Table 1. We observe energy reduction ranging from 1.23 - 4.21x for the QIF neuron models. These savings are attributed to the non-linear dynamics of the QIF neuron, which tends to induce a voltage distribution with neurons further away from the threshold, as seen in Figure 9. These dynamics increase the difficulty for a neuron to spike, reducing the rate at which less important neurons may fire due to noise or low-quality features. Additionally, neuromorphic datasets show greater energy savings on average than static datasets. This difference may stem from the high sparsity and noise typical of these datasets that cause LIF models to follow noise and produce excess spikes while the QIF models handle this noise more effectively, reducing unnecessary spikes.'
> >
> > Note that we also changed the way we present our energy efficiency results. Before, we used the percent difference equation, which measures the percent difference of two values from their mean. After internal discussion, we decided this metric may lead to confusion when interpreting our results. Therefore, we now present our energy improvements in terms of how many times more efficient is the QIF neuron model compared to the LIF neuron model. We believe this provides a more intuitive interpretation of the energy reductions.
> >
> > Additionally, at the end of Section 5.2, we add additional discussion on the computational overhead of the QIF neuron model on both neuromorphic and non-neuromorphic hardware. This discussion aims to showcase the additional overhead of the QIF neuron model we observed on non-neuromorphic hardware. We then discuss how neuromorphic hardware can potentially mitigate this overhead due to the significant decrease in spiking activity seen when using the QIF neuron model. A snippet of this section is included.
> >
> > ''To discuss potential concerns related to computational complexity, we showcase the additional latency of a QIF neuron compared to a LIF neuron. A single LIF neuron requires one multiplication and one addition while a single QIF neuron requires three additions and two multiplications. Assuming the two additions required to compute $(u - u_{rest})$ and $(u - u_c)$, from Equation 11, can be done in parallel, the QIF neuron has one addition and multiplication more than the LIF neuron, leading to roughly 2x the computational complexity. On our non-neuromorphic experimental setup, we observe that this leads to inference time overheads between 1.04-1.39x, as shown in Table 2. Due to the limited public availability of neuromorphic hardware, it is difficult to calculate the exact computational overhead incurred by these additional operations. However, we do know that many neuromorphic hardware implementations, such as Intel Loihi 2 [1], follow event-driven paradigms. This means the lower spike rate of the QIF neuron has the potential to lower the computational overhead we observed on non-neuromorphic hardware. To put this in perspective, across all datasets and models, the QIF neuron produces an average of 45.47\% less spiking activity than the LIF neuron. Therefore, the QIF neuron will only require around half the number of active neurons during inference on average. This suggests that implementing these models on neuromorphic hardware can offset the additional computational complexity of the QIF neuron through its decreased spiking activity.'
> >
> > Thank you again for your comments. If you have any additional questions, comments, or concerns please let us know and we will address them as soon as possible.
> >
> > [1] Intel. Taking neuromorphic computing to the next level with Loihi 2 technology brief. https://www.intel.com/content/www/us/en/research/ neuromorphic-computing-loihi-2-technology-brief.html, 2021. Accessed: 03-19-2024.

---

> > > ### Comment · Reviewer_tkRP · 2024-11-22
> > >
> > > Thank you for your effort in presenting this work. While I acknowledge the potential of introducing the QIF neuron into spiking neural network training, I believe the novelty of this contribution should be further clarified and supported. The QIF neuron itself is a well-established model in computational neuroscience, and its application in neural networks, although interesting, might not constitute a significant innovation by itself. For instance, the act of incorporating mature neuron models like Hodgkin-Huxley (HH) or Izhikevich neurons into network training has been explored previously and is often seen as a demonstration of applicability rather than a groundbreaking contribution.
> > >
> > > Moreover, the QIF model is known to be highly sensitive to hyperparameter settings, which can significantly impact its performance. To strengthen the paper, I strongly recommend including a comprehensive experimental study to systematically evaluate the sensitivity of the QIF neuron to its hyperparameters across various datasets and tasks.

---

> > > > ### Author Response · Authors · 2024-12-02
> > > >
> > > > Dear Reviewer tkRP,
> > > >
> > > > Thank you for your thoughtful and constructive feedback.
> > > >
> > > > Regarding the novelty of our contribution. While the QIF neuron is indeed a well-established model in computational neuroscience, our work goes beyond just incorporation and demonstration. In this work, we focus on tackling the difficulties that come with applying the QIF neuron model to deep SNNs such as discretization of the QIF neuron model fit for deep learning, deriving and proving a theoretical equation for choosing an appropriate surrogate gradient window, and providing extensive comparisons in the energy-efficiency, performance, convergence, and hyperparameter robustness of the QIF vs. LIF neuron models, rather than simply applying the QIF by replacing LIF neurons in existing architectures. Additionally, while we have seen previous work incorporating mature neuron models into network training, to the best of our knowledge, these works are limited to small networks that typically incur large latencies (timesteps).
> > > >
> > > > In response to your concern about hyperparameter sensitivity, we agree that the QIF neuron model can be highly sensitive to hyperparameter initialization. However, we have conducted two experimental studies to evaluate the performance of QIF neurons under varying hyperparameter settings across multiple datasets. These results are included in the revised manuscript in Appendix F. These results showcase that while performance can vary drastically based on hyperparameters, there are large ranges of parameters that lead to stable training regardless of model architecture. This is something we observed as we were able to use the same QIF neuron model hyperparameters for all training tasks and model architectures within our work. We believe these results showcase the hyperparameter robustness of the QIF neuron model and would love to discuss any further ideas you may have to better showcase this.
> > > >
> > > > Thank you again for your valuable feedback.

---

### Official Review · Reviewer_2Bpj · 2024-11-04

**Soundness:** 2
**Presentation:** 3
**Contribution:** 2
**Rating:** 3
**Confidence:** 4

**Summary:**

The author proposes a targeted neuron modeling approach to address existing issues in Spiking Neural Networks (SNNs). This approach reduces the energy consumption of Leaky Integrate-and-Fire (LIF) neurons to some extent and also facilitates faster model convergence. The author conducted validation experiments on both static and dynamic datasets and achieved successful results. This type of Quadratic Integrate-and-Fire (QIF) model could potentially become a widely applied neuron modeling paradigm.

**Strengths:**

1.  A new neuron modeling approach is proposed, along with an appropriate surrogate gradient window.
2.	The effectiveness of this method is demonstrated across multiple datasets.
3.	Evidence is provided that the QIF model can reduce energy consumption and facilitate model convergence.

**Weaknesses:**

1.	The generalizability of QIF has not been verified. QIF has only been tested on CNN-based models, and it remains unproven whether this model has good generalization to other types of SNNs.
2.	Although visualizations suggest that QIF can guide models to converge quickly, this conclusion is derived post hoc. Is there any theoretical basis for this?

**Questions:**

see weakness

---

> ### Author Response · Authors · 2024-11-21
>
> Hello Reviewer 2Bpj,
>
> We appreciate your time and effort in reviewing our work. Based on your insightful comments, we have modified the work and discussed our changes below.
>
> To address your concerns about the generalizability of the QIF neuron model, we have added two additional experiments to our work in Appendix E. The first experiment involves training a ResNet-34 model using CIFAR-10, scaling the images to $64\times64$. ResNet-34 has around $2 \times$ the parameters as ResNet-19. In this experiment, we observe a substantial energy improvement from the QIF neuron model of $2.15 \times$ less than the LIF neuron model and competitive accuracy between both models. We hope this experiment better showcases the generalizability of the QIF neuron to larger CNN architectures. In our second experiment, we use a Spiking Vision Transformer model and the Tiny ImageNet dataset to showcase the QIF neuron model on a non-CNN architecture and more complex task. We choose the Tiny ImageNet Dataset over ImageNet due to the time constraints of the rebuttal process. However, we feel that Tiny ImageNet provides useful insight due to its increased complexity and similarity to ImageNet. We are still in the process of collecting results with this model and dataset but would like to share our preliminary findings. The QIF model consumes approximately $8.20 mJ$ while the LIF neuron model consumes $.99\times$ less at $8.14 mJ$. In terms of accuracy, we initially see the LIF model outperform the QIF model. The QIF model catches back up and surpasses the LIF model later in training obtaining validation accuracies of $39.16$\% and $38.16$\% respectively. These accuracies are rather low as we only used two timesteps and did not use augmentations such as cutmix/mixup or label smoothing. We used two timesteps due to the computational cost of larger timesteps and the short rebuttal period. We didn't use additional augmentations so we can better attribute the differences in performance to the neuron model rather than the augmentations. Figures showcasing the training process and spike rates of each model are included in Appendix E. We hope his experiment showcases the QIF neuron model's applicability to other model architectures as well as more complex datasets. It remains an interesting research direction for us to explore the applicability of the QIF neuron model to tasks other than image classification as well as other network architectures.
>
> To address your concern about the lack of a theoretical basis for our results, we add additional discussions to section 5. In Section 5.2, we add the following snippet,
>
> ''Using Equation 17, we calculate the energy consumption of each neuron model in Table 1. We observe energy reduction ranging from 1.23 - 4.21x for the QIF neuron models. These savings are attributed to the non-linear dynamics of the QIF neuron, which tends to induce a voltage distribution with neurons further away from the threshold, as seen in Figure 9. These dynamics increase the difficulty for a neuron to spike, reducing the rate at which less important neurons may fire due to noise or low-quality features. Additionally, neuromorphic datasets show greater energy savings on average than static datasets. This difference may stem from the high sparsity and noise typical of these datasets that cause LIF models to follow noise and produce excess spikes while the QIF models handle this noise more effectively, reducing unnecessary spikes.'
>
> and in Section 5.4,
>
> ''Figure 15, in Appendix H, showcases that the loss landscape of the model trained with QIF neurons is considerably broader than that of the model trained with LIF neurons. The QIF model also features a broader local minimum and a smoother loss surface, which can lead to quicker model convergence and higher performance. As discussed in Section 5.2, these non-linear dynamics make it harder for a QIF neuron to fire. We believe these allow the QIF neuron model to filter out noise more effectively, enabling quicker learning of the most relevant features, and leading to faster convergence compared to the LIF neuron.'
>
> These discussions attempt to provide more understanding of how the theoretical dynamics of the QIF and LIF neuron models can both reduce energy consumption and guide models to converge quickly. We also back up our claims with a visualization of the voltage distribution found within a model. We hope these additional discussions help shed light on how the QIF neuron model can converge quicker as well as exactly why it is more energy efficient.
>
> Thank you again for your comments. If you have any additional questions, comments, or concerns please let us know and we will address them as soon as possible.

---

> > ### Author Response · Authors · 2024-11-21
> >
> > As a side note, we would like to mention that we changed the way we compare the energy efficiency of the QIF and LIF neuron models. Previously, we had used the percent difference formula, which measures the percent two values differ from their mean. However, after internal discussion, we feel this metric is not the best way to present our results as it may be interpreted incorrectly. Instead, we compare energy efficiency in terms of how many times more energy efficient the QIF neuron model is compared to the LIF neuron model. These changes are reflected throughout the paper and in Table 1.

---

### Note · Authors · 2025-01-22

I have read and agree with the venue's withdrawal policy on behalf of myself and my co-authors.